# Micron-resolution fiber mapping in histology independent of sample preparation

Marios Georgiadis [1] ✉, Franca auf der Heiden [2], Hamed Abbasi [3,4], Loes Ettema[3], Jeffrey Nirschl [5], Hossein Moein Taghavi[1], Moe Wakatsuki[1], Andy Liu[1], William Hai Dang Ho[1], Mackenzie Carlson[1,6], Michail Doukas [7], Sjors A. Koppes [7], Stijn Keereweer [4], Raymond A. Sobel [5], Kawin Setsompop[1], Congyu Liao [1], Katrin Amunts [2,8], Markus Axer [2,9], Michael Zeineh [1] & Miriam Menzel [2,3] ✉

Mapping the brain's fiber network is crucial for understanding its function and malfunction, but resolving nerve trajectories over large fields of view is challenging. Here, we show that computational scattered light imaging (ComSLI) can map fiber networks in histology independent of sample preparation, also in formalin-fixed paraffin-embedded (FFPE) tissues including whole human brain sections. We showcase this method in new and archived, animal and human brain sections, for different sample preparations (in paraffin, deparaffinized, various stains, unstained fresh-frozen). We convert microscopic orientations to microstructure-informed fiber orientation distributions (μFODs). Adapting tractography tools from diffusion magnetic resonance imaging (dMRI), we trace axonal trajectories revealing white and gray matter connectivity. These allow us to identify altered microstructure or deficient tracts in demyelinating or neurodegenerating pathology, and to show key advantages over dMRI, polarization microscopy, and structure tensor analysis. Finally, we map fibers in non-brain tissues, including muscle, bone, and blood vessels, unveiling the tissue's function. Our cost-effective, versatile approach enables micron-resolution studies of intricate fiber networks across tissues, species, diseases, and sample preparations, offering new dimensions to neuroscientific and biomedical research.

The brain contains billions of nerve fibers that form a highly complex and densely interwoven network. Detailed mapping of the neuronal trajectories is a major goal of neuroscience[1,2]. However, disentangling the fiber architecture with micrometer resolution and over large fields of view remains prohibitive: diffusion magnetic resonance imaging (dMRI) maps nerve fiber pathways in whole human brains in vivo with millimeter resolution, and even with long post-mortem scan times it still probes hundreds of variably-oriented axons per voxel[3,4]. Electron

[1]Department of Radiology, Stanford University, Stanford, CA, USA. [2]Institute of Neuroscience and Medicine (INM-1), Forschungszentrum Jülich GmbH, Jülich, Germany. [3]Department of Imaging Physics, Faculty of Applied Sciences, Delft University of Technology, Delft, the Netherlands. [4]Department of Otorhinolaryngology and Head and Neck Surgery, Erasmus MC, University Medical Center Rotterdam, Rotterdam, the Netherlands. [5]Department of Pathology, Stanford University, Stanford, CA, USA. [6]Department of Neurology and Neurological Sciences, Stanford University, Stanford, CA, USA. [7]Department of Pathology, Erasmus MC, University Medical Center Rotterdam, Rotterdam, the Netherlands. [8]C. and O. Vogt Institute for Brain Research, University Hospital Düsseldorf, Medical Faculty, University, Düsseldorf, Germany. [9]Department of Physics, School of Mathematics and Natural Sciences, University of Wuppertal, Wuppertal, Germany. ✉e-mail: mariosg@stanford.edu; m.menzel@tudelft.nl

microscopy resolves nerve cells at nanometer resolution[5,6], but tracking the fiber pathways is limited to small tissue volumes[7,8]. Small-angle X-ray scattering (SAXS) can directly probe multi-directional nerve fiber pathways[9–11], yet it requires expensive equipment such as synchrotrons and raster-scanning the sample with a pencil beam makes it time-consuming. Small-angle light scattering (SALS) similarly uses a pencil beam of visible light photons to provide local fiber orientations[12,13], but cannot reach micrometer resolution and requires long scans and dedicated laser illumination and detection equipment[12]. Various light microscopy techniques visualize nerve fibers at high resolution and over larger fields of view, using staining[14,15], fluorescence[16], or other contrast mechanisms. However, they rely on structure tensor analysis to derive fiber orientations[17,18], which decreases effective resolution and is limited in resolving crossing or densely packed nerve fibers. Polarization-based techniques directly visualize fiber orientations exploiting their birefringence[19,20], with 3D-polarized light imaging (3D-PLI) being the current gold standard for analyzing nerve fiber architecture in whole human brain sections[21–23]. In spite of this, 3D-PLI analysis yields only unidirectional fiber orientations per pixel, and tissue preparation needs to preserve birefringence, e.g., by avoiding dehydration and lipid removal. Recently, scattered light imaging (SLI) has emerged as a method to reveal multi-directional, densely interwoven nerve fiber networks in unstained brain cryo-sections with micrometer resolution, using a stationary LED matrix display and a camera to probe the anisotropic scattering of the sample[24–26].

At the same time, the most common tissue preparation method in histology is paraffin embedding and microtome sectioning, followed by histological staining[27]. These formalin-fixed paraffin-embedded (FFPE) sections, which are abundantly available in hospital pathology archives as well as in research and clinical laboratories, cannot be analyzed by polarization-based techniques because the birefringence is significantly reduced during tissue preparation. One prominent example is the *BigBrain* atlas[14], containing thousands of FFPE sections of a whole human brain. Nissl-based structure tensor analysis (Nissl-ST) can be used to track white matter fibers in FFPE brain sections by analyzing the arrangement of glial cells to derive nerve fiber directions[28]. However, Nissl-ST can only be applied to sections with Nissl staining. Furthermore, it has an effective resolution of tens of micrometers, and similar to polarization microscopy, it does not resolve multi-directional, crossing fibers per pixel, which are abundant in the brain.

Here, we show that computational scattered light imaging (ComSLI) – a fast and cost-effective optical microscopy method that uses a directed rotating LED light and a high-resolution camera for improved signal-to-noise ratio – can reveal microscopic, multi-directional fiber orientations of virtually any new and archived tissue section, including FFPE and fresh-frozen sections. Using ComSLI, we recover fiber orientation maps from animal and human brain tissues prepared with different protocols, stains, thicknesses, and storage periods – including whole-brain sections from the *BigBrain* atlas[14,29]. We leverage the micron-resolution fiber orientation information to obtain ground-truth, microstructure-derived fiber orientation distributions (µFODs) over a range of resolutions, and use those to perform fiber tractography, resulting in microstructure-informed tractograms reflecting axonal trajectories. We show that our technique reconstructs fiber microarchitecture in healthy and diseased tissue – discerning myelin from axonal loss, revealing microstructural alterations in multiple sclerosis and leukoencephalopathy, and identifying degeneration of the perforant pathway in sclerotic and Alzheimer Disease hippocampus tissue. By comparing ComSLI to dMRI, 3D-PLI, and Nissl-ST, we demonstrate its key advantages in revealing highly detailed and interwoven fiber networks, including micron-resolution nerve fiber architecture in whole-brain FFPE sections. Finally, we show that our technique can also be applied to reveal directed structures in non-brain tissues, like muscle and collagen fibers.

Our approach enables retrospective investigations of histology slides, combining information from any stain with fiber orientations. Furthermore, it can help reveal new micro-organization patterns as well as relationships between the microstructure and the underlying biology in healthy and diseased, animal and human tissues.

## Results

### Revealing microscopic fiber architecture in whole human brain sections

Spatially oriented structures such as nerve fibers scatter light predominantly perpendicular to their main axis[30], yielding characteristic light intensity profiles, $I(\varphi)$, where the mid-position of a peak pair indicates the fiber orientation (Fig. 1A). ComSLI reverses the light path, which is physically equivalent, to determine the orientations of fibers contained in each micrometer-sized image pixel. A rotating LED light source illuminates the tissue section at ~45° elevation, and a high-resolution camera with small-acceptance-angle lens captures sample images at multiple light source rotation angles $\varphi$ (Fig. 1B). This yields an image series in which each pixel contains a light intensity profile across different illumination angles, $I(\varphi)$, according to the orientation of its contained fibers (Supplementary Movie 1). Analyzing the peak positions in each pixel's intensity profile yields fiber orientations, with example intensity profiles for single and two crossing fiber populations shown in Fig. 1G, with the signal also visualized as polar plot (see insets).

First, we tested if ComSLI can retrieve fiber orientations with micrometer resolution from extended field-of-view sections, such as entire FFPE human brain sections from the second *BigBrain* dataset[29] (Fig. 1C, inset at top right). Orientation information was recovered from single and crossing fiber bundles in each pixel, in white and gray matter (Fig. 1C). Microscopic details become apparent: Corpus callosum fibers decussate between the hemispheres with small angular variations (Fig. 1D, small arrows), while fornix fibers course superomedially, and multiple fibers cross in dense corona radiata white matter (Fig. 1E). U-fibers connect the precentral to neighboring gyri (Fig. 1F), with subjacent deeper fibers crossing the main tracts (small arrows).

Being able to detect multiple fiber orientations per pixel at micrometer scale, we then sought to study their distribution across scales. Namely, this can help to better understand the fiber distribution in MRI voxels, which are hundreds of micrometers to millimeters in size and typically contain thousands of axons. Figure 1H shows the microstructure-derived fiber orientation distributions (µFODs) for the highlighted points in the corpus callosum, corona radiata, and U-fibers (squares in Fig. 1D–F) in the form of polar histograms, for a range of different scales. For instance, in the corpus callosum, a grid of 7 × 7 pixels (corresponding to ~50 µm resolution) shows high horizontal alignment, whereas at 500 µm (71 × 71 pixels), there are two fiber populations 10° apart, which would be misinterpreted by dMRI as single fiber population due to its ~40°–45° maximum sensitivity for discerning crossing fibers[31,32]. In contrast, at 1 mm, the different fiber populations cannot be distinguished anymore.

Such multi-scale fiber distributions are rich in information. For instance, one can calculate how many brain pixels contain fiber crossings at the ComSLI resolution (Fig. 1I), and at multiple other resolutions common for MRI scanning (Supplementary Fig. 1). This reveals that only ~7% of pixels at the original 7 µm resolution contain two or more crossing in-plane fiber populations, but this rises to 87% and 95% of pixels containing microscopic crossings for 500 µm and 1 mm pixel resolution, respectively (Supplementary Fig. 1A). Conversely, the number of apparent, detectable crossings decreases at lower resolutions (Supplementary Fig. 1B), possibly as a result of the averaging of the signal and loss of orientation information at larger pixel sizes.

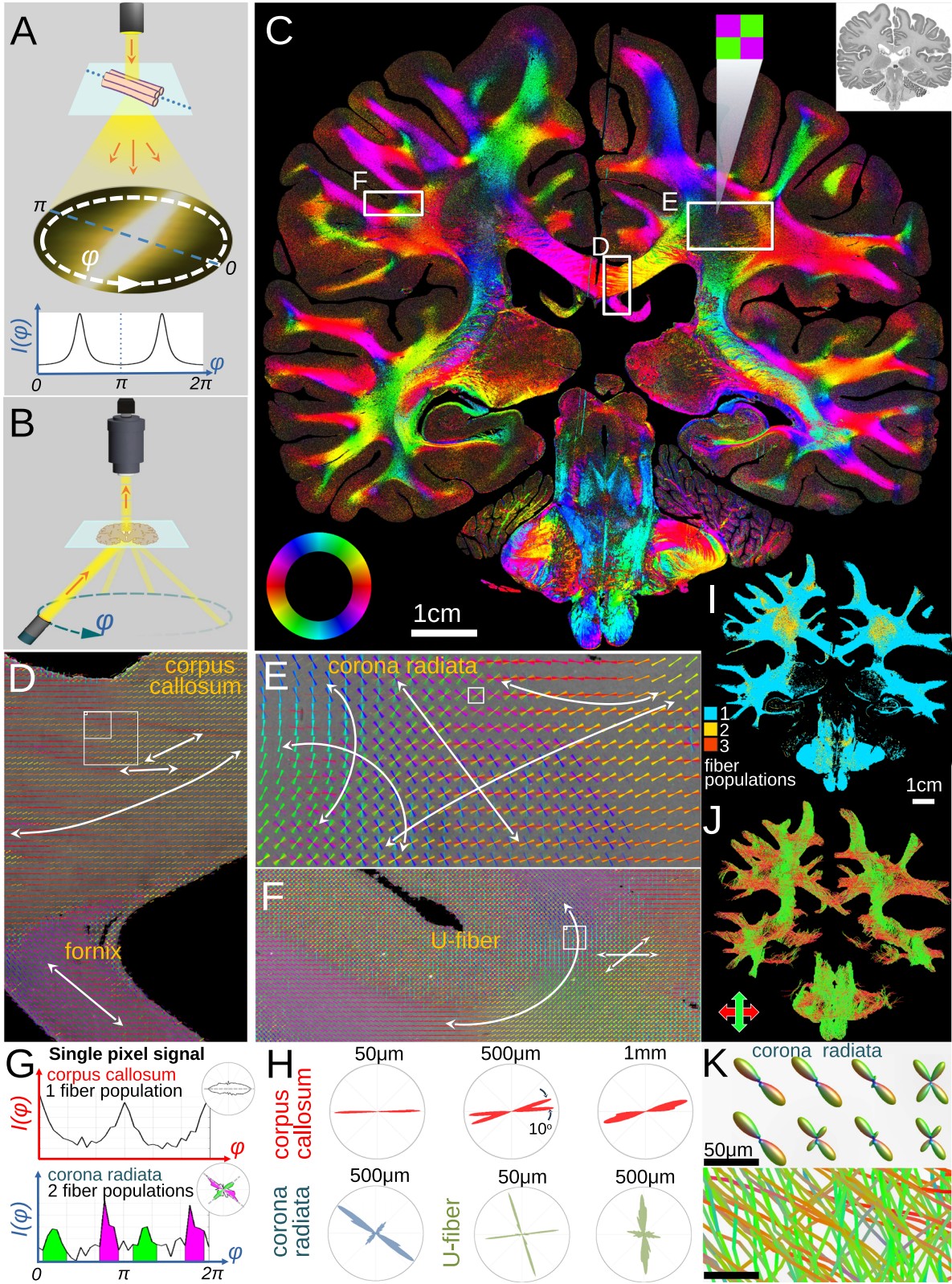

Finally, the fiber orientation distributions also enable tractography: Adapting tools employed in MR tractography, we generated orientation distribution functions (ODFs) uniquely informed by our micrometer-level fiber orientations (see Methods), and used these to produce whole-brain white matter tractograms (Fig. 1J), with microscopic ODFs and tract lines for a corona radiata region shown in Fig. 1K. The ODF and tractography analysis enable a detailed study of microstructure-based connectivity in white and gray matter of the whole brain (Supplementary Fig. 2A), and in specific anatomic regions, with some examples shown in Supplementary Fig. 2B-E, such as white and gray matter of the precentral gyrus, a pre-/post-central gyrus U-fiber, the corona radiata, and the corpus callosum. The data and code to generate the whole brain tractogram are publicly available, see Data and Code Availability sections.

**Fig. 1 | Retrieving microscopic fiber orientations in a whole human brain section. A** A light beam impinging on a fiber bundle scatters predominantly perpendicular to the fibers, resulting in paired peaks in the azimuthal intensity profile, $I(\varphi)$, whose midline indicates the in-plane fiber orientation. **B** Schematic of ComSLI measurement setup, reversing the light path to measure each pixel's azimuthal intensity profile. The tissue section is illuminated at -45° elevation by a rotating light source while a camera captures a high-resolution image of the section at each azimuthal rotation angle $\varphi$. **C** Fiber orientation map (7 μm/pixel) of the second *BigBrain* dataset, section no. 3452, FFPE, silver-stained. The inset shows the corresponding brightfield image (1 μm/pixel). The in-plane fiber orientations are shown as multi-colored pixels according to the color wheel in the bottom left; multiple colors per pixel indicate multiple fiber orientations, such as the zoomed-in green-purple corona radiata pixel. **D–F** Fiber orientations for boxes marked in (**C**). Orientations are encoded both by the line direction and its color. **D** Corpus callosum and fornix. **E** Right corona radiata. **F** U-fiber. (For better visualization, the orientation-depicting lines of sets of 15 × 15 pixels (**D**, **F**) and 75 × 75 pixels (**E**) are overlaid.) **G** Azimuthal intensity signals from pixels with single and two in-plane crossing fiber populations, at the smallest squares in (**D**) and (**E**), respectively, are also shown as polar plots (top right insets). The polar plots were rotated by 90° to better visualize fiber orientations. **H** Microstructure-derived fiber orientation distributions (μFODs), showing the fiber directions in a polar histogram for all image pixels in the squares indicated in (**D**–**F**). **I** Map of the number of in-plane fiber populations in each 7 μm pixel of the white matter. **J** Tractogram of the entire brain, top-bottom fibers are shown in green, left-right fibers in red. **K** Corona radiata detail of microscopic orientation distribution functions (ODFs) for kernels of 7 × 7 pixels, and tracts crossing the same pixels (from the top left corner of the square in (**E**)). Source data for the graphs in (**G**) and (**H**) are provided as Source Data file.

Such investigations at whole-brain level unveil microscopic fiber orientations, enabling detailed investigation of anatomy, correlations with cellular information, and pathology.

## ComSLI works across stains, preparation protocols, and species

Next, we tested whether ComSLI can retrieve information from histological sections across stains, sample preparations, and species, since the orientation of scattered light is affected solely by the orientation of scattering fibers.

We scanned consecutive FFPE sections of a human hippocampus with various stains, colors, intensities, antibody concentrations, and counter-stains (see Methods). Figure 2 and Supplementary Fig. 3 show the brightfield images (A), fiber orientation maps (B), and zoomed-in views of the cornu ammonis (CA) and multiple fiber tracts (C). Despite the different staining protocols, targets, intensities, and counter-stains, the retrieved fiber orientations are almost identical across hippocampal sections, with small anatomical differences between sections (see difference maps and corresponding histograms in Fig. 2D).

We also investigated a celloidin-embedded section, myelin-stained ~120 years ago, from the brain collection of the Cécile and Oskar Vogt Institute for Brain Research (Fig. 2E, F). Despite the section age and the low light transmission in parts of the section (Fig. 2E), detailed nerve fiber orientations were retrieved (Fig. 2F), e.g., in the corpus callosum and the Muratoff bundle (subcallosal fasciculus) running subependymal to the caudate nucleus surface[33] (arrowhead).

This led us to ask (i) if we can also retrieve orientations from sections at different steps of sample preparation, and (ii) if the sample processing affects the method's sensitivity. We scanned two sections in 3 different steps: 1) in paraffin after sectioning, 2) after deparaffinization, and 3) after staining (Supplementary Fig. 4). Excitingly, fiber orientations could be visualized from sections still in paraffin (Supplementary Fig. 4A, B), as well as from unstained sections (Supplementary Fig. 4C, D), showing similar results as the stained sections (Supplementary Fig. 4E, F). The derivation and concordance of white and gray matter orientations across sample preparation stages highlight the wide range of histology sections that can be analyzed with ComSLI.

While FFPE sections are the gold standard in histological tissue analysis, fresh-frozen samples are also common, allowing fast diagnosis during surgery or better preservation of molecular targets. Here, we show that ComSLI can retrieve fiber orientations in human brain tissues fresh-frozen after autopsy without fixation or staining, such as human visual cortex and hippocampal sections (Supplementary Fig. 5).

To confirm that the technique works across species, we measured multiple animal brain FFPE sections with different stains, including from mouse (Fig. 2G–J) and pig (Fig. 2K–N). Tracts were retrieved with micrometer resolution, such as the intricate mouse cerebellar connectivity (Fig. 2J) and multiple crossing and U-fibers in the pig corona radiata (Fig. 2N).

## Detecting disturbed microarchitecture in pathologic tissues

To test if ComSLI can unveil disturbed microscopic fiber orientations and provide unique information in pathologic human tissue, we studied human brain samples with neuropathology.

First, we investigated the brain of a subject with multiple sclerosis – a neurological autoimmune disease damaging the myelin sheaths of nerve fibers and impairing signal transmission. We imaged a section with focal demyelination (Fig. 3A, red-contoured lesions). To fully characterize the lesion pathologically, we performed additional myelin (Luxol fast blue - LFB) and neurofilament stains in consecutive sections (Fig. 3J, L, and Supplementary Figs. 6, 7). Three lesion areas are noted based on histology (arrows 1–3 in Fig. 3A, B, J): lesion area 1 has demyelination and lower neuropil density, lesion areas 2 and 3 show demyelination only. Higher magnification of characteristic regions in lesion area 1 (demyelination and neuropil loss), lesion area 2 (demyelination), and normal white matter (region 4) are shown in Fig. 3L, for neurofilament staining.

The light scattering signal (Fig. 3B) was not affected by myelin loss (lesion areas 2 and 3), but only by lower neuropil density (lesion area 1). Interestingly, the axonal orientations could be retrieved in all regions, including extensive demyelination and neuropil loss (Fig. 3C). In regions of transition between normal white matter (nwm) and demyelinated white matter (dwm) at the edge of lesion 3 (Fig. 3D-F), the ComSLI signal intensity is preserved, and the fibers seem to continue their course despite demyelination. This highlights the robustness of the method, which retrieves axonal orientations independent of myelin, also for gray matter (gm) axons (Fig. 3D, E). In addition, ComSLI detects changes in tissue density: the neuropil loss in lesion 1 leads to lower ComSLI signal (Fig. 3B) and fiber organization (Fig. 3G, H), which is detected by ComSLI based on the remaining axons. Signal preservation despite demyelination and sensitivity to neuropil loss is shown in Fig. 3K, where the three lesions (areas 1–3) have a lower myelin histology signal than the nwm regions (areas 4–5), but only lesion 1 has a lower average scattering signal.

Next, we studied an H&E section from a necrotizing leukoencephalopathy brain (Fig. 3M), which is characterized by white matter necrosis with loss of neurons and myelin, confirmed with LFB and neurofilament stains on consecutive sections (Fig. 3R, U), showing patchy axon loss and spongiosis as well as axonal spheroids and dystrophic axons (Supplementary Fig. 6I, J). The three pathology-confirmed lesion areas (indicated by arrows and red-contoured in Fig. 3M) depict a low myelin histology signal, but lesion area 3 shows a higher average scattering signal than lesions 1–2 (Fig. 3W). Upon close pathology review, lesion area 3 has less necrotizing tissue (Fig. 3S), not initially noted by an expert pathologist but later observed due to the scattering signal preservation, confirming the method's sensitivity to tissue loss. The lesions are located within the corona radiata (cr) where multiple nerve fiber crossings occur; the fiber orientation map (Fig. 3O) reveals the intricate fiber organization in this region, with

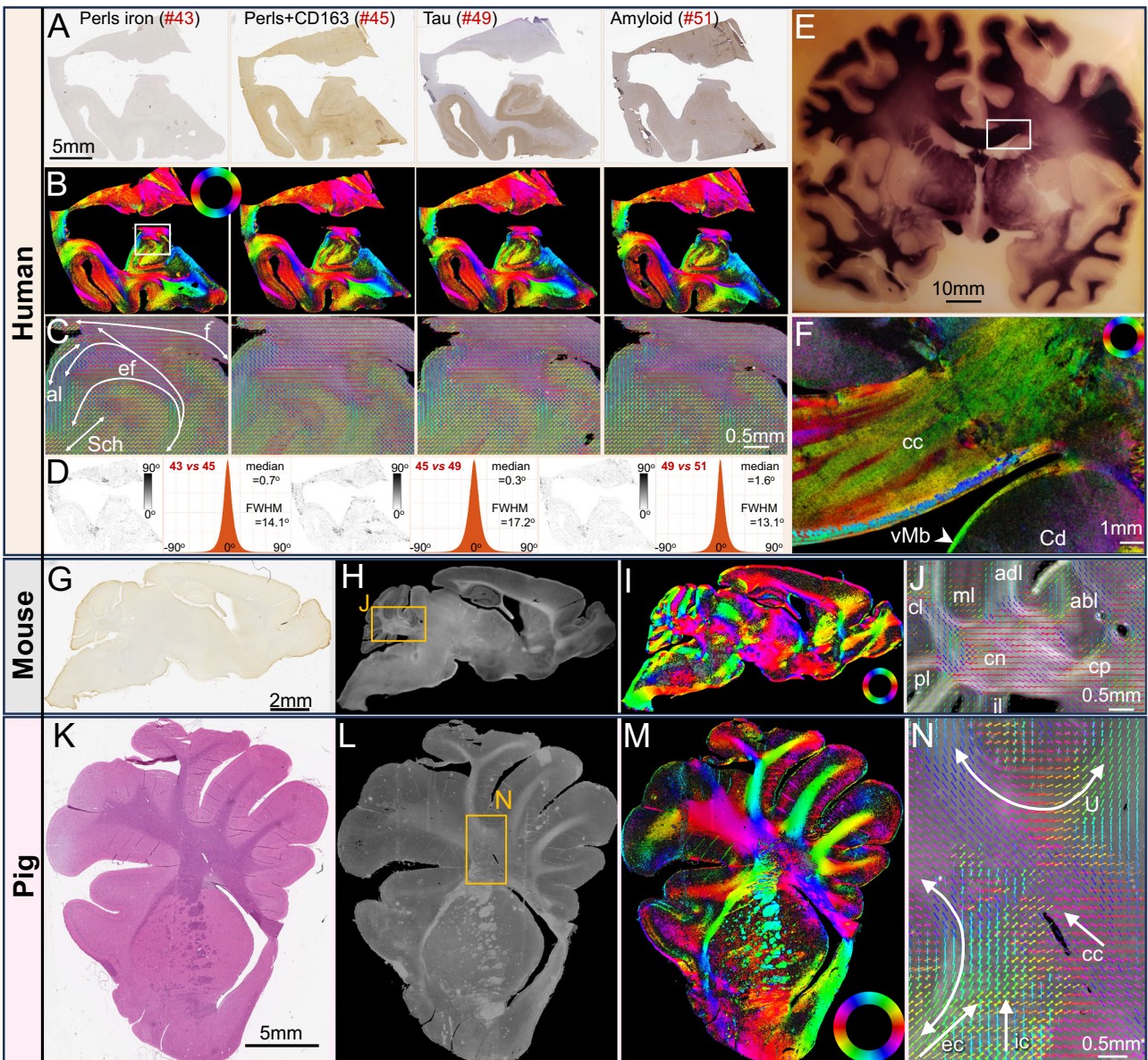

**Fig. 2 | ComSLI detects fiber orientations independent of tissue preparation and species. A–D** Human Hippocampi, FFPE sections with different stains. **A** Brightfield microscopy images (0.5 μm/pixel). **B** ComSLI fiber orientation maps (7 μm/pixel), orientations encoded by color wheel. **C** Zoomed-in fiber orientation vector maps (white box in (**B**)), including parts of cornu ammonis and multiple tracts: fornix (f), endfolial pathway (ef), alveus (al), Schaffer collaterals (Sch). Vectors of 15 × 15 pixels are overlaid. **D** Orientation difference for each pair of consecutive sections, as map (left) and histogram (right), with median value and full-width-half-maximum (FWHM) reported. Histograms were generated from the difference maps, excluding zero/background pixels (43 vs. 45: 2,850,173 pixels, 45 vs. 49: 2,784,337 pixels, 49 vs. 51: 2,787,707 pixels), with a bin size of 0.3°. **E, F** Myelin-stained human brain section (~120-year-old). **E** Section photograph. **F** ComSLI color-coded fiber orientation map (3 μm/pixel) of marked region in (**E**), cc corpus callosum, Cd caudate nucleus, vMb ventral Muratoff bundle (arrowhead). **G–J** Sagittal mouse brain section, FFPE, microglia-stained. **G** Brightfield image (0.5 μm/pixel). **H** ComSLI average scattering signal (4.25 μm/pixel). **I** Fiber orientation map. **J** Zoom-in of box in (**H**) (cerebellar area), including cerebellar peduncle (cp), cerebellar nuclei (cn), lobes (inferior-il, posterior-pl, central-cl, anterodorsal-adl, anterobasal-abl), and gray matter including oriented molecular layer (ml) fibers. Vectors overlaid for 15 × 15 pixels. **K–N** Pig hemisphere-brain section, FFPE, H&E-stained. **K** Brightfield image (0.5 μm/pixel). **L** ComSLI average scattering signal (9 μm/pixel). **M** Fiber orientation map. **N** Fiber orientations from box in (**L**), overlaid for 15 × 15 pixels, including U-fiber tracts (curved arrows) and multiple crossing fibers (straight arrows); external capsule (ec), internal capsule (ic), corpus callosum (cc). Source data for the histograms in (**D**) are provided as Source Data file.

grayish-appearing color showing crossing fibers of multiple orientations. Figure 3Q, T show multiple fiber crossings inside and around the lesions, with fibers more heterogeneously distributed inside the lesions (example heterogeneous orientations are highlighted by the arrowheads), reflecting the disorganization associated with local tissue destruction. This is also quantitatively reflected in the μFODs shown in Fig. 3X, where the lesions show relatively preserved but more heterogeneous distributions than the corresponding perilesional normal white matter areas (exact areas are depicted in Fig. 3V). For instance, lesion area 1 seems to lose most of the horizontal fibers (orange arrows in Fig. 3Q), and lesion area 2 most of the two minor diagonal fiber populations (orange arrows in Fig. 3T).

We next focused on the hippocampus, a key region for memory, affected in neurodegeneration. We compared hippocampal pathways in coronal sections of samples from 3 donors: one with epilepsy-associated hippocampal sclerosis, one with Alzheimer Disease (AD),

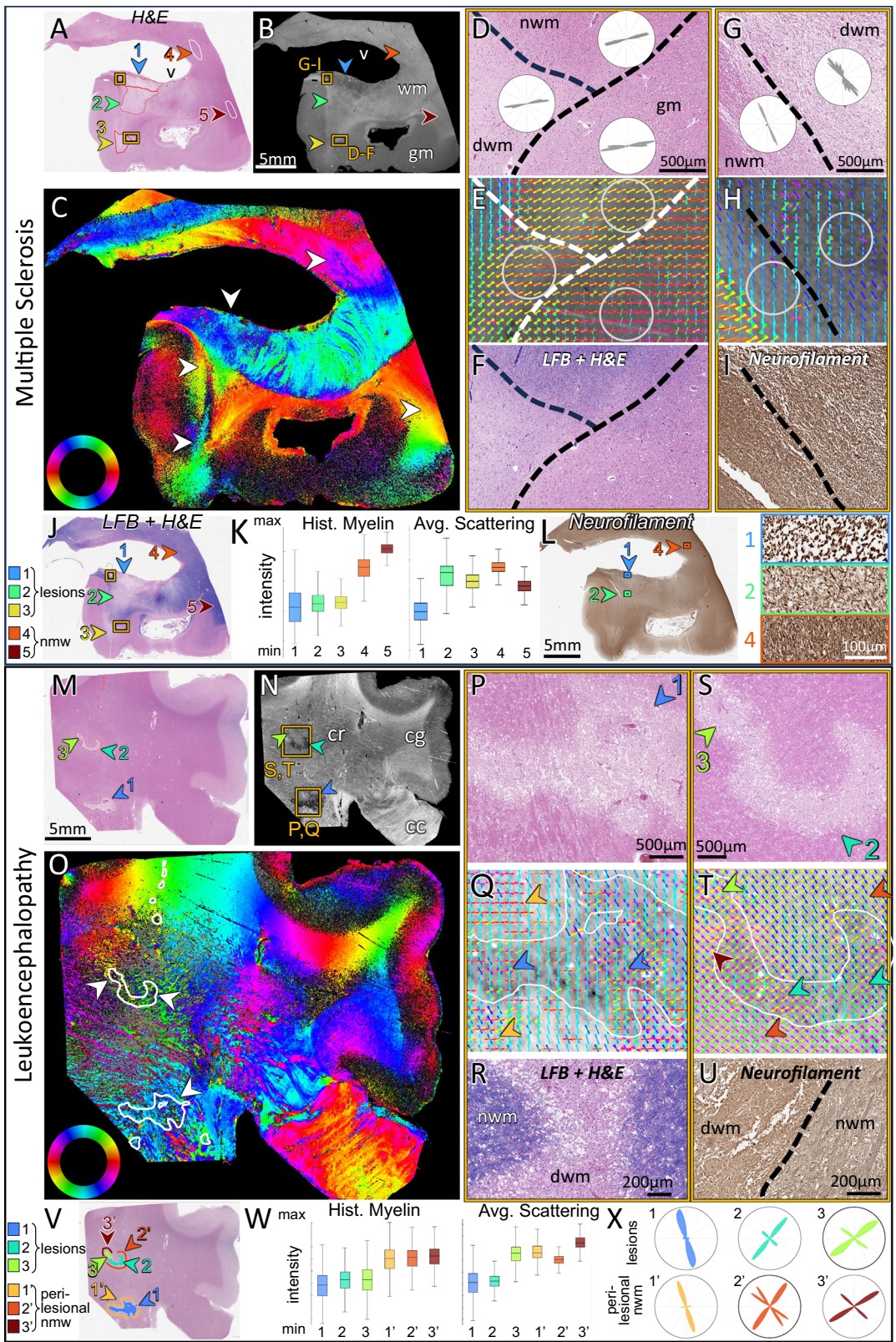

and an aged control, Supplementary Fig. 8. At a macroscopic level, the diminished volume of the sclerotic and AD hippocampi, Supplementary Fig. 8B', B", was accompanied by strong pathway degeneration, with few remaining crossings across the sclerotic hippocampus and a greatly reduced number in the AD hippocampus compared to the control, Supplementary Fig. 8C, C', C". The perforant pathway (p), the principal source of entorhinal input to the hippocampus, overlying the medial subiculum, was almost completely degenerated in the sclerotic and was very weak in the AD hippocampus, as shown with fiber orientations and resulting tractograms, Supplementary Fig. 8D'–F' and D"–F". On the other hand, the control hippocampus had multiple interwoven fibers across all subfields, Supplementary Fig. 8C, with an especially strong perforant pathway connecting the entorhinal cortex to CA1, CA3, and dentate gyrus, Supplementary Fig. 8D–F. Overall,

**Fig. 3 | Human brain pathology measured with ComSLI (FFPE sections, H&E-/LFB-/neurofilament-stained).** Brightfield microscopy: 0.5 μm/pixel, ComSLI: 9 μm/pixel. **A–L** Multiple sclerosis brain. **A** Brightfield H&E image with lesion areas 1–3 (red contours) and normal white matter areas 4–5 (white contours). **B** Average scattering signal (v: ventricle, gm: gray matter, wm: white matter). **C** Fiber orientation map. **D** Zoom-in of H&E-stained lesion 3 (lower box in (**A**), (**B**)), including normal white matter (nwm), demyelinated white matter (dwm), and gray matter (gm), with fiber distributions from underlying areas, also circled in (**E**). **E** Corresponding fiber orientations. 15 × 15 pixel orientations are overlaid for visual clarity. **F** The same region in a consecutive LFB + H&E-stained section. **G** Zoom-in of H&E-stained lesion 1 (upper box in (**A**), (**B**)), with fiber distributions from underlying areas, also circled in (**H**). **H** Corresponding fiber orientations. **I** The same region in a consecutive neurofilament-stained section. **J** LFB + H&E-stained consecutive section, with lesion and nwm areas depicted. **K** Intensity of histology myelin stain (left) and average ComSLI scattering (right) for areas 1–5. Box plots were generated from all pixels contained in the respective regions [1, 2, 3, 4, 5], $n$ = [1.33, 1.35, 0.56, 0.66, 0.68]*$10^5$ pixels. Histology box plot: range = [0–157, 14–121, 29–109, 71–183, 119–186], quartiles = [37–85, 54–81, 59–79, 113–141, 144–161], median = [61, 67, 69, 128, 153] intensity units. Scattering box plot: range = [30–60, 44–78, 47–70, 57–73, 48–65], quartiles = [42–49, 57–66, 56–62, 63–67, 55–59], median = [45, 63, 59, 65, 57] intensity units. **L** Neurofilament-stained consecutive section and zoom-ins of areas 1,2 and 4. **M–X** Leukoencephalopathy brain. **M** Brightfield H&E image with lesion areas 1-3 (red contours). **N** Average scattering signal. cc: corpus callosum, cg: cingulum, cr: corona radiata. **O** Fiber orientation map, lesions with white contour. **P** Zoom-in of H&E-stained lesion 1 (lower box in N). **Q** Corresponding fiber orientations, more aligned outside and less within the lesion (orange/blue arrows respectively). **R** Lesion zoom-in in LFB + H&E-stained consecutive section, showing dramatic myelin loss, axonal loss and spongiosis in lesion. **S** Zoom-in of H&E-stained lesions 2–3 (upper box in N). **T** Corresponding fiber orientations, with arrows highlighting aligned (perilesional and lesion 3) and heterogeneous (lesion 2) fiber orientations. **U** Lesion zoom-in in a neurofilament-stained consecutive section. **V** Lesions and nwm areas depicted. **W** Intensity of histology myelin stain (left) and average ComSLI scattering (right) for the three lesions (1–3) and their perilesional areas (1′–3′). Box plots were generated from all pixels contained in the respective regions [1,2,3,1′,2′,3′], $n$ = [4.87, 2.01, 0.78, 3.40, 1.35, 0.81]*$10^4$ pixels. Histology box plot: range = [8–139, 27–134, 15–146, 53–176, 64–168, 70–169], quartiles = [54–88, 67–94, 64–97, 99–130, 103–129, 107–132], median = [72, 81, 81, 116, 118, 120]. Scattering box plot: range = [79–111, 88–105, 96–119, 99–118, 100–111, 105–120], quartiles = [91–99, 94–98, 105–110, 106–111, 104–107, 110–114], median = [96, 96, 108, 108, 105, 112] intensity units. **X** Fiber orientation distributions for the lesions and their perilesional nwm areas. Source data for the box plots in (**K**) and (**W**), and the fiber orientation distributions in (**D**), (**G**), and (**X**) are provided as Source Data file.

ComSLI can reveal loss of distributed axonal networks associated with neurodegeneration.

## Comparison to diffusion MRI, structure tensor analysis, and 3D-PLI

We compared ComSLI to previously developed methods that yield fiber orientations using different mechanisms and physical phenomena.

We first contrasted ComSLI to Nissl-based Structure Tensor (Nissl-ST)[28]. Nissl-ST uses 2D structure tensor analysis to retrieve fiber orientation information in white matter areas based on the anisotropic shape and alignment of Nissl-stained glial cells. We measured brain sections Nissl-stained with Cresyl violet (Fig. 4 and Supplementary Fig. 12), using the code and settings by Schurr & Mezer[28] to derive in-plane fiber orientations. Comparing Nissl-ST output for different kernel sizes, ranging from 200 × 200 μm² to 50 × 50 μm² (Supplementary Fig. 9), the 100 μm kernel showed the best balance between data quality and resolution, and was used in our analyses.

Contrasting ComSLI results on a whole human brain section (Fig. 4, left) to those obtained using Nissl-ST on the same section (Fig. 4, middle), the fiber orientations are in overall agreement between the two methods (Fig. 4A, A′). However, ComSLI achieves better visualization of small features in a label-free manner, such as fiber bundles in the putamen (Pu) and globus pallidus (GPe, GPi), the medial and lateral medullary laminae (mml, lml), and the external capsule (ec), Fig. 4B, B′. ComSLI also better resolves, e.g., the two fiber populations in the corpus callosum (Fig. 4C, C′). Moreover, ComSLI resolves multiple crossing fibers per 3 μm pixel, for example, in the corona radiata region (Fig. 4D), with Nissl-ST providing single fiber orientation per 100 μm pixel (Fig. 4D′). This is exemplified by the fiber orientation distributions as shown in the insets, whereby Nissl-ST often misinterprets crossing fibers as single fibers at an intermediate angle.

Direct quantitative pixel-wise comparison between the two methods (Supplementary Fig. 10) reflects the aforementioned observations. The angle differences are skewed towards 0° (Supplementary Fig. 10C, D), with small differences in the corpus callosum, but widespread angle differences in crossing fiber areas such as the corona radiata (Supplementary Fig. 10B). This is also shown via the generated ComSLI ODFs in a corpus callosum and corona radiata area (Supplementary Fig. 10E and F), where differences in pixel size (boxes), information content, and orientation accuracy become evident (the blue lines in the dashed boxes indicate Nissl-ST fiber orientations). This exemplifies ComSLI's ability to discern finer orientations, which also translates to microstructure-informed fiber tracts in simple and complex crossing fiber areas (Supplementary Fig. 10G, H). Gray matter comparison was not pursued given Nissl-ST's focus on white matter[28].

We also compared our results to fiber orientations from a high-resolution in vivo 18-h diffusion MRI scan of a healthy volunteer's brain[34] (Fig. 4, right). ComSLI fiber orientations were compared to those from the MR plane best matching the measured brain sections (see Methods). The main nerve fiber orientation patterns of ComSLI and dMRI are in agreement (Fig. 4A, A″). However, ComSLI unveils distinct microscopic fiber orientation patterns, like the different fiber populations at low angles in the corpus callosum (Fig. 4C), which are bundled in a single fiber orientation in dMRI (Fig. 4C″), or the individual fiber bundles in the putamen and globus pallidus or the medullary laminae (Fig. 4B), not resolved by dMRI in vivo (Fig. 4B″) or ex vivo[35].

A pixel-wise, quantitative comparison of ComSLI and Nissl-ST on a different brain section – the cell-body silver-stained *BigBrain* section from Fig. 1C – and comparison to in-vivo dMRI orientations from a corresponding plane yields similar conclusions (Supplementary Fig. 11).

Finally, 3D-Polarized Light Imaging (3D-PLI) provides micron-resolution fiber orientation maps of whole-brain sections[21-23]. Our tests on the studied whole-brain FFPE sections showed significantly reduced sensitivity (Supplementary Fig. 12, right), compared to the high-sensitivity high-resolution maps it provides in 3D-PLI sections. This is possibly because an important part of myelin lipids resolves during alcohol dehydration[36], and water removal also distorts myelin periodicity[37], resulting in loss of its birefringence.

## Mapping fibers in histology beyond brain

Finally, we tested whether ComSLI can retrieve fiber orientations from non-brain human tissues, given the multiple fiber types present, often in close proximity. We performed measurements on FFPE sections from different tissue types (Fig. 5): human tongue muscle, colorectal tissue, lower jaw bone (mandible), and arterial wall.

In the tongue, ComSLI reveals layers of differently oriented structures (Fig. 5A–D), like outer epithelium (1), underlying stroma (2) with collagen fibers running mostly parallel to the surface, and inner muscular layer (3), which contains muscle fibers that enable tongue reshaping and multi-directional movement.

In the colorectal tissue (Fig. 5E–I), orientations of multiple fiber types are retrieved: epithelial lamina propria (1) is mostly oriented normally/radially to the surface, while basal membrane fibers are mostly oriented parallel to the surface (arrowheads in Fig. 5G); the mucosal and submucosal fibers (2) have a complex architecture with various orientations; circular muscle fibers of the muscularis mucosa

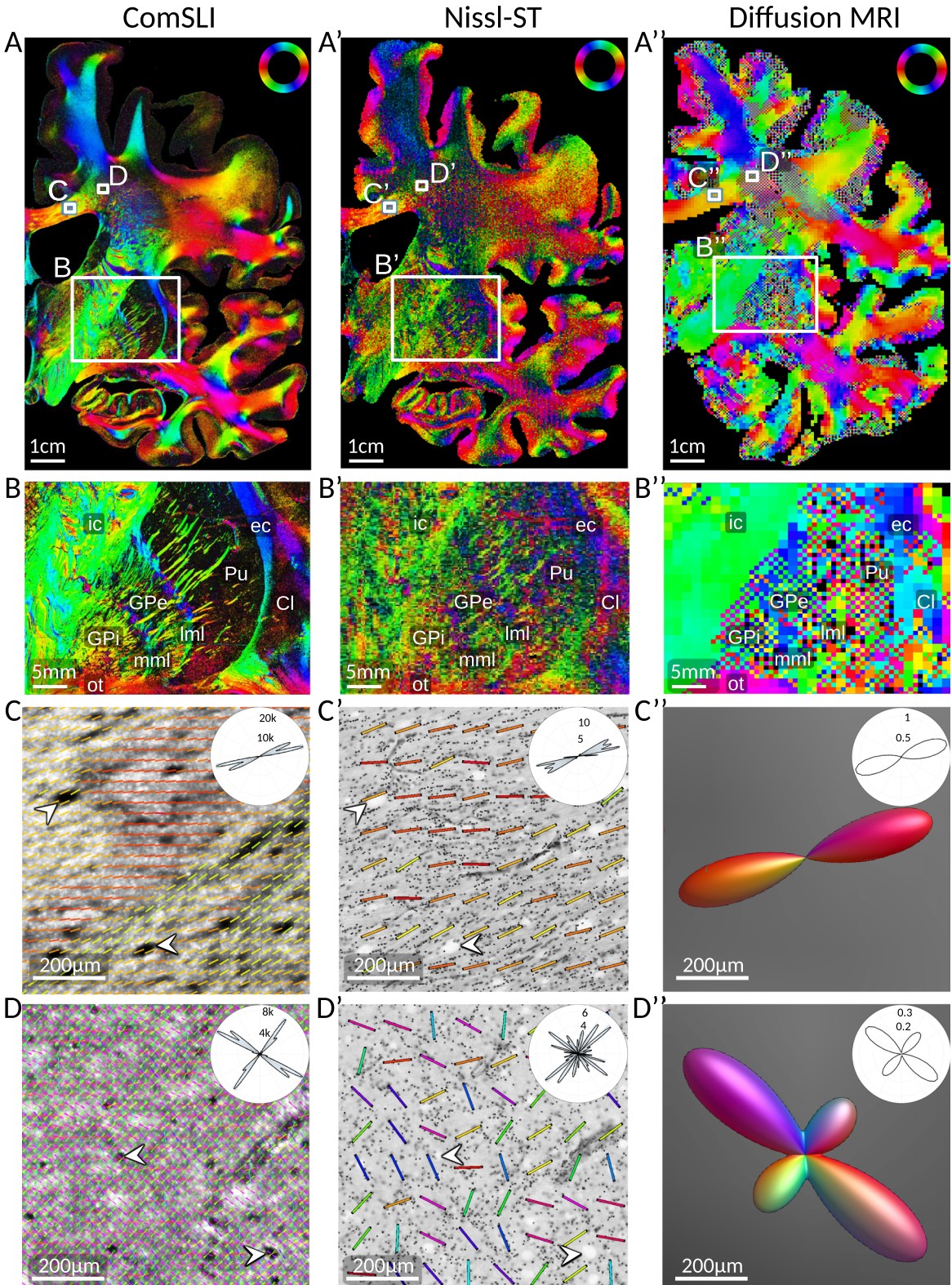

(3) run from top to bottom (corresponding inset in Fig. 5G and lower arrowhead in Fig. 5H); longitudinal fibers of the muscularis propria (4) run mostly perpendicular to the circular muscle fibers (corresponding inset in Fig. 5G and upper arrowhead in Fig. 5H).

Blood vessels yield a high scattering signal (arrowheads in Fig. 5F), with collagen fibers wrapping around vessel walls (arrowheads in Fig. 5I).

In the mandible bone (Fig. 5J–L), the inner spongy (/trabecular) bone (2) shows fibers mostly following trabecular microstructure[38] (white arrowheads). The U-shaped cortical bone (1) shows a dominant fiber orientation in the direction of the force applied during mastication, i.e., from top left to bottom right (see corresponding insets 1a-c in Fig. 5L), while the superior region (3) exhibits more variable fiber orientations, distributing incoming loads to the rest of the jaw.

**Fig. 4 | Comparison of ComSLI, Nissl-ST, and dMRI fiber orientations.** ComSLI and Nissl-ST were performed on the same brain section (human coronal FFPE section, Nissl-stained with Cresyl violet, no. 3301) – the ComSLI measurement with 3 μm pixel size, and Nissl-ST on the brightfield microscopy image with 1 μm pixel size. Diffusion MRI was performed in vivo on a healthy volunteer with 0.76 mm isotropic voxel size, and the orientation distribution functions were evaluated at a plane similar to the evaluated brain section (see Methods). **A** Color-coded fiber orientation is depicted for each image pixel. The orientations were computed with *SLIX*[72] for ComSLI and dMRI, and with the provided code for Nissl-ST[28] using a 15 μm blur radius and a kernel size of 100 × 100 μm². **B** Enlarged views of the fiber orientation maps for the bottom rectangular area marked in (**A**), featuring individual fiber bundles in the globus pallidus and putamen; annotated are internal capsule (ic), external capsule (ec), optic tract (ot), globus pallidus external segment (GPe), globus pallidus internal segment (GPi), putamen (Pu), medial and lateral medullary laminae (mml, lml), and claustrum (Cl). **C** Enlarged views of the fiber orientations for the corpus callosum area corresponding to the size of a dMRI voxel (760 μm), annotated by top left rectangle in (**A**). ComSLI fiber orientations were visualized as colored lines and overlaid on 10 × 10 pixels for better visualization; white arrows indicate the same microscopic features in ComSLI average scattering signal in (**C**) and histology in (**C'**). Fiber orientation distributions for each modality are shown as insets in the top right. **D** Enlarged views of the fiber orientations for the corona radiata area corresponding to the size of a dMRI voxel (760 μm), annotated by top right rectangle in (**A**). ComSLI fiber orientations were visualized as colored lines and overlaid on 10 × 10 pixels for better visualization; white arrows indicate the same microscopic features in ComSLI average scattering signal in (**D**) and histology in (**D'**). Fiber orientation distributions for each modality are shown as insets in the top right. The dMRI orientation distribution functions in (**C''**) and (**D''**) were visualized with *MRtrix3*'s *mrview*[41]. Source data for the fiber orientation distributions in (**C**) and (**D**) are provided as Source Data file.

In the artery wall (Fig. 5M−O), ComSLI reveals microscopic arrangements of fibers in different layers. While the brightfield image (Fig. 5M) indicates the fiber type (pink area dominated by collagen fibers, dark/black area by elastin fibers, gray-green area by muscle fibers, indicated by arrowheads in corresponding color), the fiber orientation map (Fig. 5O) reveals distinct fiber directions in each layer, critical for artery mechanical properties[39]: fibers run mostly parallel to the surface in the inner (1), middle (3), and outer (5) layers; in the layers in between, the fibers run mostly (2) or also (4) perpendicular to the surface (see corresponding insets).

As in brain tissue, ComSLI reveals directed structures of non-brain tissues both for stained and unstained sections (Supplementary Fig. 13), and yields similar results across adjacent sections, for different section thicknesses (from 4 μm to 15 μm) and different steps of sample preparation, also for sections still in paraffin (Supplementary Fig. 13A, D, G).

## Discussion

Nerve fibers form a dense network in the brain, determining its connectivity and function in health and disease. Here, we show that ComSLI retrieves microscopic fiber orientations across different species, diseases, and tissue types, in new and archived histology sections that were prepared with various stains and protocols at different steps of sample preparation. Such information would otherwise be undetectable with brightfield microscopy and inaccessible with other methods.

To enable a detailed study of fiber trajectories, we analyzed the derived micron-resolution orientation maps. This included calculating the number of crossing fibers in the brain (Supplementary Fig. 1). When studying the presence of microscopic crossings at multiple resolutions, only 7% of 7 μm white matter pixels showed in-plane crossings, which rises to ~95% of 1 mm pixels containing microscopic crossings (Supplementary Fig. 1A). This is higher than previous estimates[31], highlighting the fact that lower resolutions underestimate the number of crossings that are actually present and can be revealed with higher (microscopic in our case) resolutions. This effect is shown in Supplementary Fig. 1B, where we study detectability of crossings by resolution for the same pixel sizes, finding lower percentage of detectable crossings in larger pixels. This is in line with findings from a previous MRI and histology study[40], where lower resolution imaging uncovered lower number of crossings. We also calculated microstructure-derived fiber orientation distributions (μFOD), enabling quantitative analyses and providing insights about the nerve fiber architecture at multiple resolutions. Combined with powerful tools adapted from MRI tractography[41–43], this also enabled tracing axonal trajectories through the generation of orientation distribution functions (ODFs) and whole-brain and region-specific white and gray matter tractograms (e.g., Supplementary Fig. 2).

While scattered light imaging (SLI) has previously revealed interwoven nerve fiber networks with high resolution[24–26], these studies were limited to unstained, cryo- or vibratome-sections where nerve fiber architecture could also be visualized with polarization microscopy, albeit with one fiber direction per pixel. In ComSLI, we use a directed rotating high-power LED spot (instead of an LED matrix display) for improved signal-to-noise ratio, allowing mapping of microscopic fiber architectures in various types of histological samples, including formalin-fixed paraffin-embedded (FFPE) and whole human brain sections like the *BigBrain* atlas (cf. Fig. 1). FFPE sections are commonly used in research and clinical laboratories, but their microscopic nerve fiber architecture cannot be analyzed with polarization microscopy due to a lack of birefringence (Supplementary Fig. 12).

Although ComSLI can provide fiber orientations in white and gray matter at different steps of sample preparation, it seems to have low sensitivity to gray matter fiber orientations in the almost transparent, deparaffinized sections (Supplementary Fig. 4). This can be explained by a combination of effects: first, paraffin has a higher refractive index than axons which can increase the scattering of (unmyelinated) fibers in the cortex. Second, treatment with xylene to remove paraffin results in an almost complete loss of refractive index differences, which is the basis of the scattered light signal[30], rendering the sample almost fully transparent (Supplementary Fig. 4C, D). Although ComSLI can still retrieve signal from the white matter, it is unable to discern gray matter fiber orientations at this specific stage due to very low signal-to-noise levels. Staining appears to partly restore the refractive index differences and enable retrieving some fiber orientations in cortical gray matter.

Though previous work only studied healthy brain samples[24–26], we demonstrate here that ComSLI can provide novel information and valuable insights in neuropathology. By studying samples with diseases such as multiple sclerosis and leukoencephalopathy, we show that ComSLI, being robust to demyelination, can detect disturbed microarchitecture changes in lesions, and can also discern myelin from axonal loss, helping to differentiate lesion areas (Fig. 3). Moreover, taking advantage of the tractography analysis, ComSLI can reveal the loss of vulnerable pathways in degenerating brains, such as perforant fibers in a sclerotic and Alzheimer Disease hippocampus (Supplementary Fig. 8). Thus, ComSLI can be applied to the study of multiple neurologic and psychiatric disorders with structural connectivity changes, such as Alzheimer disease[44], Parkinson disease[45], traumatic brain injury[46], stroke[47], multiple sclerosis[48], schizophrenia[49], epilepsy[50], or autism[51]. It can be used across any brain circuitry and can validate ex vivo diffusion MRI orientations[4], especially in areas with small or highly interwoven fiber tracts such as the hippocampus[23] or U-fibers[52], where sensitivity and high-resolution is key. Finally, it can be used to create reference maps of microscopic fiber architecture for entire brains, e.g., applied to and extending datasets such as the *BigBrain* atlas[14,29].

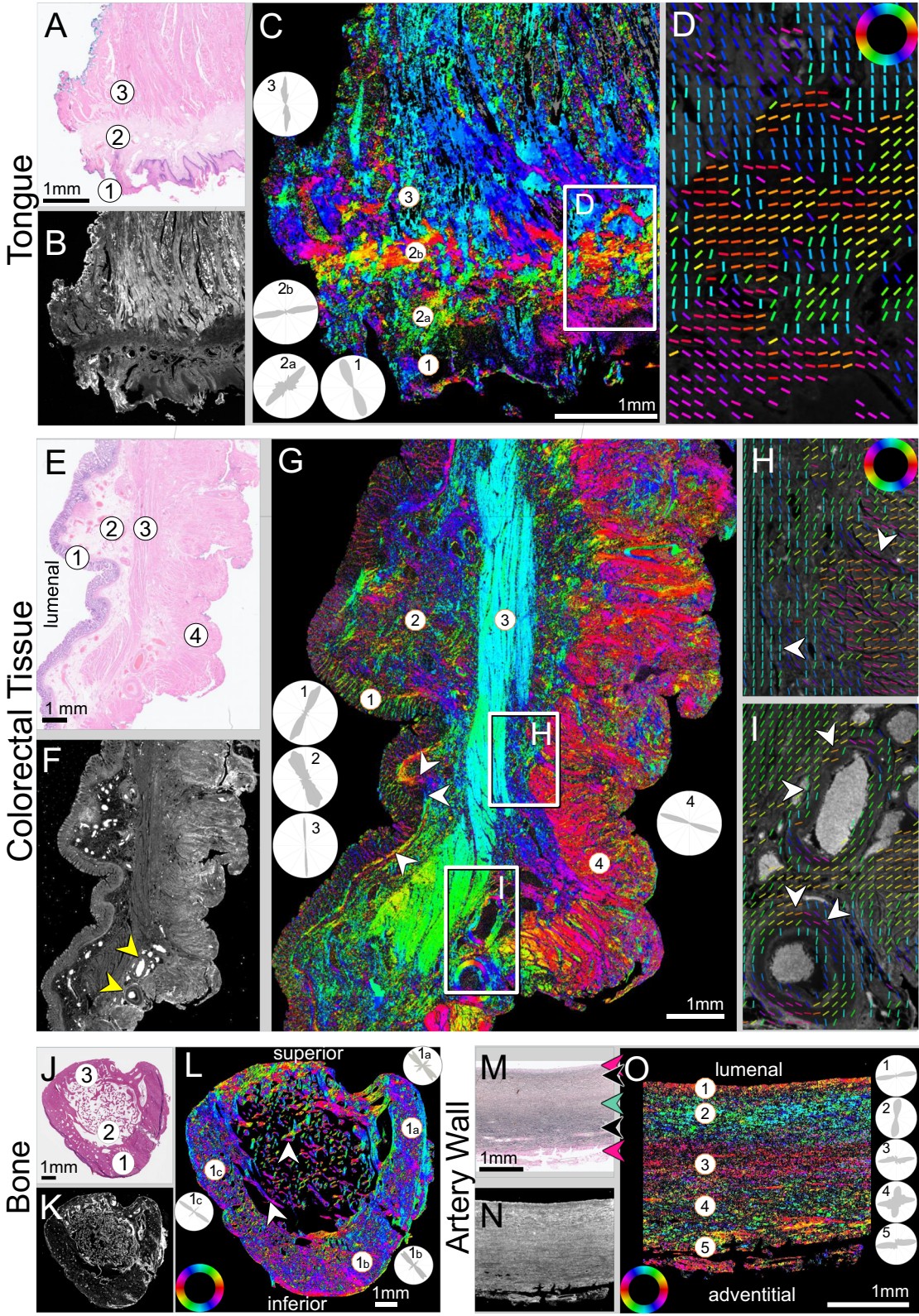

Finally, we applied our technique to non-brain samples – including muscle, bone and blood vessel tissues (Fig. 5 and Supplementary Fig. 13): ComSLI reveals not only nerve fiber architecture, but also other directed structures like collagen, muscle, and elastin fibers, which form intricate biological fiber networks that control mechanical and functional properties, and are altered in disease in muscle[53], connective[54–56], and epithelial[39,57] tissues. Our findings highlight the wide applicability of ComSLI, which can be used to reveal collagen[58],

actin/myosin[59], elastin[60], or the (usually collagenous) fiber organization around tumors[61,62], which is related to staging and malignancy of the tumor, potentially allowing more accurate diagnosis and tailored cancer treatment.

The presented approach has the potential to make fiber mapping accessible to every laboratory: ComSLI is i) relatively low-cost, requiring only an LED spot, a sample stage, and a micron-resolution camera, ii) fast, with a full measurement performed within seconds, iii)

**Fig. 5 | Non-brain samples measured with ComSLI. A–D** Human tongue muscle. **A** Brightfield H&E image (0.46 µm/pixel); (1) epithelium, (2) stroma, (3) inner muscle layer. **B** Average scattering signal (3 µm/pixel). **C** ComSLI fiber orientation map, color-coded according to the color wheel in (**D**). The insets show the fiber orientation distributions for the marked circular areas. **D** Zoomed-in area from (**C**) including different fiber layers; fiber orientations are displayed as colored lines for every 25th pixel. **E–I** Human colorectal tissue. **E** Brightfield H&E image (0.46 µm/pixel); (1) epithelium, (2) mucosa and submucosa, (3) circular muscle, (4) longitudinal muscle. **F** Average scattering signal (3 µm/pixel), with two blood vessels indicated by arrowheads. **G** ComSLI fiber orientation map, color-coded according to the color wheel in (**H**). The insets show the fiber orientation distributions for the marked circular areas. Arrowheads indicate the basal membrane. **H, I** Upper and lower zoomed-in areas from (**G**); fiber orientations are displayed as colored lines for every 40th image pixel. Arrowheads indicate vertical and horizontal fiber orientations from circular and longitudinal muscle, respectively (**H**), and blood vessels with fibers running circumferentially (**I**). **J–L** Human mandible. **J** Brightfield H&E image (0.46 µm/pixel); (1) cortical bone on anterior side, (2) trabecular (cancellous) bone, (3) cortical bone on tooth side. **K** Average scattering signal (3 µm/pixel). **L** ComSLI fiber orientation map, color-coded according to the color wheel. The insets show the fiber orientation distributions for the marked circular areas in the cortical bone. The arrowheads indicate spongy bone fiber orientations following the microstructure. **M–O** Elastic artery wall. **M** Brightfield elastin-stained image (0.46 µm/pixel). **N** Average scattering signal (3 µm/pixel). **O** ComSLI fiber orientation map. Highlighted are fiber layers with different directions going from lumenal to adventitial surfaces. The insets show the fiber orientation distributions for the entire layers at the position of the marked circular areas. All samples are 4 µm-thin FFPE sections and stained with either H&E (tongue, colorectal, mandible) or elastin staining (artery wall). Source data for the fiber orientation distributions in (**C**), (**G**), (**L**) and (**O**) are provided as Source Data file.

compatible with sections prepared using virtually any protocol. In miniaturized form, using an LED-ring instead of a rotating LED, ComSLI can be an add-on to existing microscopes and easily be integrated into clinical workflows. This enables microscopic exploration of fiber orientations in health and disease on sections readily available in thousands of laboratories and clinics worldwide.

Regarding alternative approaches, we attempted to compare ComSLI to those that can provide fiber orientations for similar samples and scales. Structure tensor analysis was recently presented for reconstructing nerve fiber pathways in FFPE Nissl-stained brain sections (Nissl-ST)[28]. Compared to Nissl-ST, ComSLI provides significantly higher resolution in a label-free manner, including multiple crossing fiber orientations per image pixel (Fig. 4, Supplementary Figs. 10 and 11C), and the possibility for out-of-plane orientation information[24–26]. Diffusion MRI can image living human subjects tomographically but lacks cellular and microscopic details, which can help better define tracts and solve key tractography problems[63]. Compared to diffusion MRI, ComSLI provides much higher resolution leading to visualization of small fiber tracts invisible to MRI (Fig. 4 and Supplementary Fig. 11).

Our study has shown that ComSLI can be applied to various kinds of histological sections: FFPE, new or archived – even century-old – at different sample preparation stages, with various stains or section thicknesses – even to unstained sections still in paraffin (Supplementary Fig. 4), or unfixed fresh-frozen sections (Supplementary Fig. 5). Together with previously shown applications on fixed cryo-sections[24] or vibratome sections[26], our study reveals the potential of ComSLI to reconstruct fiber organization in virtually any histology section in clinics or laboratories.

Given its independence from myelin birefringence, ComSLI can yield neuropathologic insights when combined with myelin-sensitive microscopic methods such as 3D-PLI[21–23], polarization-sensitive optical coherence tomography[19,20], or X-ray scattering[9–11,38,64]. Compared to 3D-PLI, which is the method-of-choice for micron-resolution fiber mapping on sections, ComSLI can be applied to various kinds of histological sections independent of sample preparation or birefringence, including FFPE sections, and can resolve crossing fibers per pixel, with an easy-to-implement and cost-effective setup.

Scattering of light to study fiber orientations has also been previously used in methods such as small-angle light scattering (SALS), which captures light scattering from tissues by raster-scanning them using a pencil laser beam of hundreds of micrometers[12,65]. In contrast, ComSLI captures a centimeter-wide field of view within seconds with micrometer resolution and without the need for specialized laser beam equipment. Similar to SALS, we have previously applied coherent Fourier scatterometry using a laser beam and capturing light at both low and wide angles to study the physics of light scattering from brain tissues[66], and we found excellent agreement of the overall fiber

orientations with those determined by scattered light imaging in the same tissue sections[24].

As a light microscopy transmission-based technique, ComSLI analyzes histological tissue sections, and the tissue thickness might affect the measurement results: with a higher tissue thickness, there are more fiber layers that scatter light (increasing the signal), but also higher light attenuation (decreasing the signal). In previous studies, through-scattered light signals could be retrieved from sections as thick as 500 µm using laser sources[12]. Here, we tried to assess the effect of tissue thickness in multiple ways: First, indirectly, by studying multiple commonly used tissue thicknesses, e.g., FFPE sections ranging from 4 µm (all hippocampal and pathological human sections) to 20 µm thick for the (cell-body stained) *BigBrain* sections. Second, we tried to directly assess the effect of thickness: As shown in Supplementary Fig. 13, it is possible to accurately and reproducibly reveal fiber orientations across different section thicknesses (demonstrated for tongue muscle tissue of 4–15 µm thickness). For unstained sections without paraffin embedding, brain tissue sections of 60 µm thickness have been successfully analyzed[24,25], as well as vibratome sections of approximately 100 µm thickness. As for thicker samples, or samples that cannot be sectioned, future work will investigate the realization of back-scattered light imaging (instead of transmission mode as shown here) to probe tissue surfaces independent of tissue thickness.

Our study has some limitations. We only show in-plane fiber orientations, but the method also provides information of fiber inclination[24–26], and our future work will focus on quantifying out-of-plane angles. As light can scatter off any structure, imperfections on the microscope slide or coverslip (e.g., scratches or dust) will introduce artifacts. To this end, the slide should be cleaned (e.g., using wet cotton tips) before measurement, while unremovable scratches will introduce artifacts in the image that need to be excluded from analysis. Regarding tractography applications, the low section thickness only allows to follow underlying anatomic tracts over a short course, limiting the study of single sections to short-range fiber tracts; analysis of consecutive sections of a 3D specimen can enable studying longer-range tracts and will be the topic of next studies. Also, the resulting tractograms should be interpreted with caution, as the employed tractography methods were developed for dMRI which has a lower resolution than ComSLI, and the microscopic resolution can result in fiber streamlines "jumping" between neighboring tracts with small angle separations; refinement of tractography algorithms for ComSLI and detailed anatomical validation using tracer studies will be the subject of future work. In addition, ComSLI provides fiber orientations without distinguishing between nerve, muscle, collagen, or other types of fibers. Future studies will investigate the different scattering signatures from distinct fiber types to enable label-free, multiplexed, fiber-specific analyses.

In conclusion, we have shown that ComSLI reveals the underlying fiber microarchitecture of tissues, including paraffin-embedded

formalin-fixed (FFPE) sections, independent of staining and birefringence, making it unique in revealing micron-resolution nerve fiber architecture in FFPE sections, which are commonly used in laboratories worldwide. This fast, simple-to-implement, and cost-effective technique can become a routine tool to provide micrometer fiber orientation mapping of histological sections, opening new avenues in the investigation of the microarchitecture of tissues in health and disease.

## Methods

### Sample preparation

**Whole human brain (BigBrain) sections.** The silver-stained human brain section (Fig. 1, Supplementary Movie 1, and Supplementary Figs. 1, 2, and 11) was obtained from a 30-year-old, male body donor without neurological disorders. The brain was removed within 24 hours after death, fixed in 4% formalin, dehydrated in increasing alcohol series (80%, 90%, 96%, 100% ethanol for at least one week), and embedded in 57–60 °C paraffin solution for two to three months. Subsequently, the brain was coronally cut into 20-μm-thin sections from anterior to posterior using a large-scale microtome (Leica SM2500 Microtome) and then mounted. The sections were placed in a decreasing alcohol series to remove the paraffin, stained with silver following the protocol of Merker[67] to highlight neuronal cell bodies, and mounted on glass slides. The silver staining represents a modified Gallyas staining introduced by Zilles[68]. It differs from the original Merker staining in that the concentration of hydrogen peroxide in the formic solution is higher and that sections are rinsed in 0.5% acetic acid twice for 5 min just before the section is immersed into the physical developer, resulting in a higher contrast of neuronal cell bodies in paraffin sections. The sections are part of the so-called second *Big-Brain* dataset[29], 3D-reconstructed with the same spatial resolution of 20 μm isotropic such as the original Jülich *BigBrain*[14]. For our study, we used section no. 3452.

The Cresyl violet-stained human brain sections (Fig. 4, and Supplementary Figs. 9, 10, and 12) were obtained from a 71-year-old, male body donor without neurological disorders. The brain was prepared as described above, but stained with Cresyl violet instead of silver. While Cresyl violet stains the rough endoplasmic reticulum, silver staining is labeling the cytoskeleton. Both are equivalent in its histological selectivity. For our study, sections no. 3301 (Fig. 4, and Supplementary Figs. 9 and 10) and 2520 (Supplementary Fig. 12) were selected and measured one and a half years after tissue preparation.

**120-year-old myelin-stained human brain section.** The myelin-stained human brain section (Fig. 2B) comes from the brain collection of the Cécile and Oskar Vogt Institute for Brain Research, Heinrich Heine University Düsseldorf, Germany. The brain of a 25-year-old male was embedded in celloidin and stained in the Vogt lab at that time according to a modified Weigert's iron hematoxylin myelin staining in 1904. The myeloarchitectonic staining has been performed as illustrated in Vogt (1903)[69], although the precise protocols are not delivered.

**Human hippocampus, cortex, and pathology FFPE brain sections.** Four-millimeter thick formalin-fixed human specimens were dehydrated in increasing ethanol steps (70% x2, 95% x2, 100% x3, 3.5 h each step), cleared in xylene (3.5 h x2), paraffin-embedded (3.5 h x2), and sectioned into 5-μm-thin sections. The sections were de-waxed and stained with agents as indicated. The hippocampal sections in Fig. 2A and Supplementary Fig. 3 were from an 89-year-old male with AD pathology, stained against microglia (CD163), Perls iron with Diaminobenzidine (DAB) enhancement, tau, and amyloid, with hematoxylin counterstain where indicated[70]. Sections from brains with multiple sclerosis (80 years old, male, from temporal periventricular white matter and cortex) and leukoencephalopathy (43 years old, male, from periventricular white matter and cingulum) were stained with hematoxylin and eosin, Luxol fast blue plus hematoxylin and eosin, and for neurofilament (2F11) (Fig. 3 and Supplementary Figs. 6, 7). Hippocampal and visual cortex sections in Supplementary Fig. 4 were from a 60-year-old male stained with hematoxylin & eosin and a 67-year-old female stained with Luxol fast blue, respectively. The sclerotic hippocampal section in Supplementary Fig. 8 was from a 69-year-old female with epilepsy, the control was from a 66-year-old female with no neuropathologic abnormality, and the AD tau-stained hippocampus was the same as in Fig. 2A, as described above.

**Fresh-frozen human hippocampus and visual cortex sections.** The human hippocampus and primary visual cortex fresh-frozen sections (Supplementary Fig. 5) were from an 88-year-old male with Lewy Body Disease, low Alzheimer disease pathology, and cerebrovascular dementia. The brain was processed according to Stanford ADRC procedures; after autopsy, it was cut in 5 mm coronal slabs and frozen using frozen metal plates in dry ice. The specimens were excised from the frozen slab, and 30 μm sections were cut using a cryostat. The sections were uncoverslipped and let thaw under the microscope, which happened in the first ~100 seconds given their thickness.

**Mouse brain section.** A female ~10-week-old C57BL/6 mouse (Jackson Laboratories) was housed in a temperature-controlled environment, with a 12-hour light/dark schedule and *ad libitum* food/water access. It was euthanized for the purposes of a different study (Stanford APLAC protocol no. 32577) under anesthesia with 2–3% isoflurane followed by cardiac puncture and perfusion with 20 mL phosphate-buffered saline (PBS). The brain was harvested, kept in 4% paraformaldehyde (PFA) in PBS for 24 h at 4 °C, transferred to 10%, 20%, and 30% sucrose in PBS, embedded in Tissue-Tek O.C.T. in dry ice for 1 hour, and cut sagittally into 10 μm sections using a cryotome (Leica CM1860). The sections were subsequently washed, mounted on a glass slide, incubated with Iba1 primary antibody (dilution 1:200), secondary antibody (goat anti-rabbit Cy3 1:200), and coverslipped. A mid-sagittal section was selected for evaluation (Fig. 2G–J).

**Pig brain section.** A 4-week female Yorkshire pig (no. 2) was euthanized for a different study (Stanford APLAC protocol no. 33684), the brain was harvested, cut into 5-mm coronal slabs using a brain slicer, and a mid-frontal slab (no. 5) was paraffin-embedded, similar to the human pathologic specimen preparation above. The slab was cut in 10 μm sections using a Leica HistoCore AUTOCUT microtome. After deparaffinization, a section (no. 127) was stained with hematoxylin and eosin and coverslipped (Fig. 2K-N).

**Human tongue, colorectal, bone, and artery wall sections.** The non-brain tissue sections (Fig. 5 and Supplementary Fig. 13) were obtained from a tissue archive at the Erasmus Medical Center in Rotterdam (the Netherlands) which provides tissue samples that were collected from patients during surgery. The bone sample was decalcified first using DecalMATE by Milestone Medical. Afterwards, all samples were fixed in 4% formaldehyde for 24 h, dehydrated in an increasing alcohol series (70%, 80%, 90%, 96%, and 100% ethanol), treated with xylene, embedded in paraffin, and cut with a microtome (Leica RM2165) into 4-μm-thin sections (for the human tongue sections in Supplementary Fig. 13, an adjacent 10-μm and 15-μm-thin section was cut off). The sections were placed in a decreasing alcohol series to remove the paraffin, mounted on glass slides, stained with hematoxylin and eosin (artery wall with Verhoeff-Van Gieson elastin staining), and then coverslipped.

**Brightfield microscopy.** The whole human brain sections were scanned with the TissueScope LE120 Slide Scanner by Huron Digital Pathology, Huron Technologies International Inc. The device measures

in brightfield mode with 20X magnification and 0.74 NA, providing a pixel size of 0.4 μm. The final images were stored with a pixel size of 1 μm.

The hippocampus, cortex, pathology, and animal brain sections were scanned using an Aperio AT2 whole-slide scanner with the ImageScope software and a 20X magnification, resulting in brightfield images with a pixel size of 0.5 μm.

The stained non-brain microscopy slides were scanned using the Nanozoomer 2.0 HT digital slide scanner by Hamamatsu Photonics K.K., offering a 20X magnification and a pixel size of 0.46 μm. The unstained non-brain microscopy slides were scanned using the Keyence VHX-6000 Digital Microscope (with VH-ZST objective, 20X), with a pixel size of 10 μm.

## ComSLI

**Whole human brain (silver-stained), hippocampus, cortex, pathology, and animal brain sections.** Measurements were performed with a rotating light source and camera (cf. Fig. 1B), using a Flexacam C3 12 MP microscope camera (Leica) and a Navitar 12X Zoom Lens with a 0.67X Standard Adapter and a 0.5X Lens Attachment, with 4.25 to 9 μm pixel size, as indicated in the figure captions. As light source, an ADJ Pinspot LED II was used, with 5.1 cm diameter and 3.5° full-angle of divergence, oriented at ~45° with respect to the sample plane. A motorized specimen stage enabled the whole-human-brain section scanning in 8 × 5 tiles; all other brain sections were scanned at a single tile. Images were acquired at 10° rotation steps (36 images/sample) with 125 ms exposure time, except from the sections in paraffin that gave very strong scattering and were imaged with 7 ms exposure time. Prior to the measurement, a 100 mm diameter diffuser plate (Thorlabs) was measured for calibration (see below for calibration details).

**Whole human brain (Cresyl violet) and non-brain tissue sections.** Measurements were similarly performed with a rotating light source and camera (cf. Fig. 1B), using a fiber-coupled LED light source consisting of an ultra-high power LED (UHP-FB-W50 by Prizmatix) with 400–750 nm wavelength (peak at 443 nm), 2-meter long step-index multimode silica (low OH) fiber patch cord (Thorlabs), a 25.4 mm diameter collimating optics (FCM1-0.5-CN by Prizmatix), and a 25.4 mm diameter engineered diffuser (beam shaper) for homogenizing the illumination (ED1-S20-MD by Thorlabs), yielding top-hat beam with 20° full-angle of divergence. The exposure time was adjusted manually per sample for maximizing the dynamic range of the captured signal while avoiding saturation (range: 50–100 ms). The light source was oriented at ~45° with respect to the sample and rotated with a motorized specimen stage (ZABER X-RSB060AD-E01-KX14A) in steps of 15° (24 images/sample). Images were taken with a 20 MP monochromatic CMOS camera (BASLER acA5472–17um) and a Rodenstock Apo-Rodagon-D120 Lens, yielding a pixel size of 3 μm (4 μm optical resolution) and a field-of-view of 16 × 11 mm². A motorized specimen stage was used to perform whole-slide scanning. Prior to the measurement, a diffuser plate (DG100x100 N-BK7 ground glass diffuser, 1500 grit, Thorlabs) was measured for calibration.

**120-year-old myelin-stained human brain section.** The measurement was performed with a similar camera and lens as for the non-brain tissue sections (BASLER acA5472-17uc and Rodenstock Apo-Rodagon-D120), using an LED display instead of a focused light source (50 × 50 cm², 128 × 128 RGB-LEDs, Absen Polaris 3.9pro In/Outdoor LED Cabinet). The sample was illuminated by a green circle segment (9° azimuthal and polar widths, respectively) with an effective illumination angle of 47°, which was rotated in 15° steps. Images were taken with 10 seconds exposure time and a gain of 10, and 4 images were averaged per illumination angle to increase signal to noise.

## ComSLI image analysis

**Flat-field correction.** Prior to each measurement session, a diffuser plate was measured under similar conditions. A 100-pixel Gaussian blur was applied to diffuser images to homogenize defects. Subsequently, the blurred images of all angles were divided by the average of their maxima for normalization. These normalized diffuser images were used to calibrate the measured tissue images, aiming to account for the uneven illumination across the field of view for each image: Each tissue image was divided by its corresponding normalized diffuser image of the same illumination angle.

**Generation of fiber orientation and vector maps.** Each calibrated image series from a ComSLI measurement was evaluated with the open-source software *SLIX*[71], which analyzes the position of scattering peaks in each pixel's intensity-vs-illumination angle plot (Fig. 1G) to compute the fiber orientations and visualize them in color-encoded maps, using multi-colored pixels and colored vector lines. Measurements with 15° azimuthal steps were processed without filtering. Measurements with 10° azimuthal steps were processed with Fourier low pass filter (40% cutoff frequency, 0.225 window width) before generating the parameter maps, as described in ref. 25. Previous studies have validated the step size as well as the filtering parameters used[24,25].

**Number of crossings.** To study the number of fiber crossings, as presented in Fig. 1I and Supplementary Fig. 1, we used the output of the orientations of the *SLIX* software[71]. Briefly, *SLIX* determines the number of orientations in each microscopic pixel by finding the number of peak pairs in each pixel's intensity-vs-angle plot (Fig. 1G) that correspond to fiber populations with distinct orientations[24,71] (up to 3 fiber populations per microscopic pixel). Each peak pair indicates one fiber population, i.e., an intensity profile with one peak pair (2 peaks) corresponds to a single fiber population, two peak pairs (4 peaks) correspond to two fiber populations, and 3 peak pairs (6 peaks) to 3 fiber populations (the maximum that can robustly and accurately be resolved in a micrometer-sized pixel). For the multi-resolution analysis in Supplementary Fig. 1 A, we took into account the number of microscopic crossings in each kernel making up the lower resolution "pixels". For example, if a 7 × 7 kernel of 7-micrometer ComSLI pixels (corresponding to an area of ~50 × 50 μm²) contained microscopic pixels with two (or three) fiber populations, indicating microscopic crossings, the lower resolution "pixel" was displayed in yellow (or red), indicating that two (or three) fiber populations are present within this kernel. For the analysis of apparent crossings in Supplementary Fig. 1B, we down-sampled the original ComSLI images to the respective resolutions, and employed *SLIX* to retrieve fiber orientations per pixel in the downsampled datasets, from which the number of orientations per pixel was determined.

**Microstructure-derived fiber orientation distributions (μFODs).** To calculate and plot the μFODs, the ComSLI fiber orientation map was partitioned into pixel kernels depending on the intended μFOD resolution. For instance, for a ComSLI dataset with a pixel size of 7 μm and intended μFOD resolutions of ~50 or ~500 μm, a 7 × 7 or 71 × 71 pixel kernel was used, respectively (Fig. 1H). Similarly, to calculate μFODs for specific regions of interest, e.g., the lesions in pathology (Fig. 3), orientations of all pixels in the area were considered. All fiber orientations of all pixels within each kernel or area of interest were then rendered as a polar histogram in 20 bins of 9°, covering 0° to 180°, and mirrored to 180°–360°. To create a continuous polar function representing the μFOD, the mid-points of the bins were fitted by a spline, which was then rendered as polar plot (Figs. 1H, 3D, G, X, 4C, D, 5C, G, L, O).

**Diffusion MR-based orientation distribution functions (ODFs) and tractography.** To generate ODFs and enable tractography, we opted for using existing tools developed in MR tractography, given the advanced and widely tested algorithms used in the field. To achieve that, an artificial diffusion MRI dataset was created based on the ComSLI µFODs described above. The dataset included 3 $b = 0$ ms/µm$^2$ and 60 $b = 1$ ms/µm$^2$ volumes. The 3 $b = 0$ ms/µm$^2$ volumes had a value of 1 at all voxels. The 60 $b = 1$ ms/µm$^2$ volumes consisted of 3 sets: **i)** The first 20 were derived for the 20 in-plane orientations of the µFOD angles (20 angles, 9° apart covering 180°), see above. There, the signal for each angle was calculated for every pixel as $e^{\frac{-n}{10}}$, where $n$ is the frequency of fiber orientations for that angle in that pixel, based on the µFOD polar histogram. **ii)** The second 20 orientations were at the plane perpendicular to the plane of the section. For these, the signal was set to 1 for all pixels, indicating no signal loss and hence no axons along that plane, thus enforcing the orientations to be within the section plane. **iii)** The third set of 20 orientations was the initial set of 20 polar histogram orientations, but tilted 20° off the section plane towards the perpendicular plane. There, each pixel's signal was $1 - \frac{1 - Signal_0}{5}$, where $Signal_0$ is the signal of the first set of angles. This aimed to approximate a realistic diffusion MRI signal and fiber distribution off-plane, where the signal loss at 20° off the detected in-plane axon orientation is 20% of the signal loss along the axons, which reflects a fiber response with a sharp peak along the axis and a quick off-axis fall-off, as described by Tournier et al.[72], and is expected for the following tractography steps that include estimating the fiber response.

To compute the ODFs and tractograms, *MRtrix3*[41] functions were used with the generated artificial MRI signal along with the corresponding bval and bvec files as input: *dwi2response* with the *fa* algorithm and l$_{max}$ = 6 to calculate the fiber response and *dwi2fod* with the *csd* algorithm and l$_{max}$ = 6 to calculate the ODFs. Finally, the *tckgen* function was used to generate tractograms, with a minimum tract length of 2 mm, which was found to be reasonable to compute given the micron-scale pixel sizes.

### Nissl-ST

The fiber orientation maps were computed in Matlab following the procedure described and code shared by Schurr & Mezer[28], using default settings and 100 µm as kernel to compute the structure tensor (effective resolution).

### ComSLI – Histology registration

To enable quantitative, pixel-wise comparison between ComSLI and Nissl-ST outputs (Fig. 4 and Supplementary Figs. 11, 12) as well as quantification of the ComSLI and histology stain intensities (Fig. 3 and Supplementary Fig. 7), the histology images were linearly 2D-registered to the corresponding ComSLI average scattering images of the same section using the *antsRegistration* function in *ANTs*[73] (options: *--transform Similarity[0.1], --convergence [100 x 70 x 50 x 0,1e-6,10] --smoothing-sigmas 3x2x1x0vox --shrink-factors 8x4x2x1*).

To register consecutive histology sections (Figs. 2, 3, and Supplementary Fig. 7) the *SyN* transform of the same *ANTs* command was used (options: *--transform SyN[0.1,3,0] --convergence [800 x 400 x 200 x 100 x 70 x 50 x 0,1e-6,10] --smoothing-sigmas 15x9x5x3x2x1x0vox --shrink-factors 64x32x16x8x4x2x1*).

### 3D-PLI

The 3D-PLI measurements were performed using the LMP3D microscope (Taorad GmbH, Germany), containing an evo4070MFLGEC (2048x2048) camera and a Nikon 4x (NA 0.2) lens, which achieves a pixel size of 1.85 µm and an in-plane optical resolution of 2.2 µm (determined by US-Airforce target). The sample was illuminated by linearly polarized light at 20° rotation angles and analyzed by a circular analyzer as described by Axer et al.[22]. The 3D-PLI FOM was computed

on the supercomputer JURECA at Forschungszentrum Jülich (grant no. 28954).

### Diffusion MRI

The diffusion MRI dataset[34] is from a 30-year-old male who underwent 18 hours of diffusion MRI scanning in the MGH-USC 3 T Connectom scanner using gSlider-SMS[74]. After manually identifying the MR plane that most closely matched the *BigBrain* histology sections, the entire dataset was rotated using FreeSurfer's *freeview* and the b-vectors were rotated at the same angles (rotation angles for the Silver-stained section were −34° sagittal and 1.5° axial, for the Cresyl violet −30.4° sagittal). Fiber responses and orientation distributions were computed using the *dwi2response* and *dwi2fod* functions in *MRtrix3*[41], using the multi-tissue, multi-shell algorithm[42], and visualized in *mrview*. To generate whole-brain colormaps in Fig. 4 and Supplementary Fig. 11, *MRtrix3*'s *sh2amp* function was used to probe fiber orientations at the coronal plane at 5° intervals, and colormaps were generated using *SLIX*[71].

### Statistics & reproducibility

Statistical analysis of data was performed in form of histograms and box plots, when comparing adjacent tissue sections with different stains (histograms in Fig. 2D), when comparing ComSLI to Nissl-ST fiber orientations (histograms in Supplementary Figs. 10 and 11), or when analyzing demyelinated vs. normal myelinated regions in neuropathological tissues (box plots in Fig. 3K, W and Supplementary Fig. 7D). The histograms and box plots show the distribution of pixel values for the respective sample/region; the *n*-number (number of data points used to generate the histogram or box plot) has been indicated in the respective figure legend together with other statistical parameters like median and quartiles. The bright-field microscopy images shown in various figures (Figs. 2–5; Supplementary Figs. 3, 4, 6–8) have been obtained from measurements of distinct samples (without repetition or averaging) as these were mostly used as reference and not the focus of the current study.

Reproducibility of experimental findings was demonstrated by performing measurements of the same sample after different steps of sample preparation and by performing measurements across different species. To show that fiber organization can be reconstructed independent of sample preparation, adjacent tissue sections with different staining and with different tissue thicknesses were measured. All attempts confirm the reproducibility of results.

No statistical method was used to predetermine sample size. Sample sizes were chosen to provide sufficient evidence that ComSLI can map fibers in histological tissue sections independent of sample preparation. For this purpose, various types of samples were chosen (including adjacent tissue sections, different tissue types, tissues from different species, tissues with different pathologies, tissues with different thicknesses, and differently prepared tissues).

No data were excluded from the analysis. The experiments were not randomized as the studies were performed on existing archived tissue sections. The investigators were not blinded to allocation during experiments and outcome assessment; blind studies were not necessary for demonstrating the capabilities of the presented method.

### Ethical statement

Our research complies with all relevant ethical regulations. The body donors gave written informed consent for the general use of post-mortem tissue in this study for the aims of research and education. The whole human brain sections were acquired in accordance with a vote of the ethics committee of the medical faculty of the Heinrich Heine University Düsseldorf (Germany) under protocol no. 4863. All other human brain samples were acquired in accordance with the Stanford Alzheimer's Disease Research Center (ADRC) Institutional Review

Board (IRB), Assurance no. FWA00000935. The usage of the non-brain tissue samples was approved by the Medisch Ethische Toetsing Commissie (METC) of the Erasmus Medical Center in Rotterdam (the Netherlands) under project no. MEC-2023-0587. The mouse and pig were sacrificed for a different study, approved by the Stanford Administrative Panel on Laboratory Animal Care (APLAC) under protocol nos. 32577 (mouse) and 33684 (pig).

## Reporting summary

Further information on research design is available in the Nature Portfolio Reporting Summary linked to this article.

## Data availability

All data underlying the study, the original image files, and a sample whole-brain dataset are available in the data repository Dryad[75]: https://doi.org/10.5061/dryad.02v6wwqb2. Source data are provided with this paper.

## Code availability

The initial flat-field correction of raw ComSLI images was performed with custom MATLAB code as described in the respective section. The ComSLI fiber orientation maps were generated with the open-source software *SLIX* version 2.4.2 (https://github.com/3d-pli/SLIX/). The Nissl-ST fiber orientation maps were generated in MATLAB using code shared by Schurr & Mezer[28]. The MRI orientation distribution functions (ODFs) were computed with the open-source software *MRtrix3* which is available on GitHub (https://github.com/MRtrix3/) and visualized with the tool *mrview*. The MATLAB code to generate μFODs from ComSLI orientation maps, create a virtual diffusion MRI dataset, and generate ODFs and tractography using MR tools, is provided together with data in the Dryad repository, including a sample whole-brain dataset.

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

## Acknowledgements

We thank Markus Cremer and the laboratory team at Forschungzentrum Jülich (INM-1), Germany, for preparing the whole human brain (*BigBrain*) sections, and Philipp Schlömer for performing and processing the 3D-PLI measurement. We thank David Camarillo for his help with the pig project. This work was supported by the National Institutes of Health grant nos. R03AG083702 (M.G.), 1F31AG077905 (M.C.), R01MH092311 (K.A.), R01AG061120 (M.Z.), and R21AG072675 (M.Z.), by the European

Union's Horizon 2020 Research and Innovation Programme grant nos. 945539 and 101147319 (F.H., K.A., M.A., M.M.), by the Deutsche Forschungsgemeinschaft (DFG) Project no. 498596755 (F.H., L.E., M.M), by the Klaus Tschira Boost Fund (M.M), by the Helmholtz Association's Initiative and Networking Fund through the Helmholtz International *BigBrain* Analytics and Learning Laboratory (HIBALL) under the Helmholtz International Lab grant agreement InterLabs-0015 (M.M), by the Convergence Imaging Facility and Innovation Center (CIFIC) of TU Delft, Erasmus MC, and Erasmus UR (H.A.), by the Foundation of the American Society of Neuroradiology (M.G.), by the Alzheimer's Association Research Fellowship no. 24AARF-1241479 (M.G.), and by the New Investigator Award Program of the National Alzheimer's Coordinating Center (M.G.).

## Author contributions

M.G. and M.M. co-designed the study, developed the LED-spot setup, supervised the research, analyzed and interpreted the data, prepared the figures, and wrote the first version of the manuscript. M.G. conducted ComSLI brain measurements except for the 120-year-old and the Cresyl violet whole human brain sections, and performed quantitative analyses including the µFODs, multi-modal comparisons, MR-inspired ODFs, and tractography. F.H. optimized the measurement parameters and hardware of the LED-display setup, performed the first measurements of the FFPE whole brain sections, measured and analyzed the myelin-stained human brain section, and designed the ComSLI setup sketch. H.A. measured and analyzed the non-brain tissue samples and prepared the corresponding figures. L.E. measured and analyzed the Cresyl violet-stained, whole human brain sections. J.N. contributed to the design of the pathology study, provided anatomical analysis and interpretation of the brain pathology samples. H.M.T. helped with the Nissl-ST analysis. M.W. performed initial measurements of FFPE sections. A.L. contributed to scanning of the silver-stained whole human brain section, W.H.D.H. helped with setting up the measurements with the LED-spot, and M.C. contributed to the mouse brain study. S.A.K. provided anatomical analysis and interpretation of tongue, jar bone, and blood vessel samples, M.D. of colorectal samples, S.K. of oral tissue samples, and R.A.S. of pathological human brain samples. K.S. and C.L. helped design the comparison to dMRI, and C.L. analyzed the dMRI data set. K.A. provided the *BigBrain* sections, contributed to the anatomical content, and participated in the design of the study. M.A. and M.Z. co-designed the study and contributed to data interpretation and supervision. All authors contributed to the revision and editing of the manuscript.

## Competing interests

The authors declare no competing interests.
