## [Transparent Peer Review file · Nature Communications]

Micron-resolution fiber mapping in histology independent of sample preparation

Corresponding Author: Dr Miriam Menzel

Version 1:

Reviewer comments:

Reviewer #1

(Remarks to the Author)

Georgiadis and colleagues present a method for measuring axonal (and other fibers) orientations based on SLI with micrometer resolution. Their method uses a novel setup to improve upon existing SLI-based methods and provide orientation data from different tissue preparations, including formalin-fixed paraffin-embedded sections. In addition, they provide an interesting usage of microstructure-informed orientation to create fiber orientation distributions (ODFs) similar to those used in the field of tractography. Importantly, they provide solid evidence that this new method is useful for studying microstructural changes in different pathological cases, and provide important comparisons with existing methods, showing the superiority of ComSLI compared with existing techniques. The fact that this technique is relatively cheap and easy to implement across various preparation procedures, both for stained and unstained samples, will make it of high interest to the neuroscientific and neuroimaging community.

The manuscript is well-written and I strongly recommend its publication. However, I believe that the following minor points should be addressed prior to publication.

- The authors analyzed the prevalence of crossing fibers as a function of spatial resolution. Please highlight in this section that you are referring to in-plane orientations.

- The same analysis shows that the number of crossing fibers increases with increasing "voxel" size (Supp. Fig. 1). This contrasts with previous work by Schilling et al. ("Can increased spatial resolution solve the crossing fiber problem for diffusion MRI?", NMR Biomed, 2017) that showed the opposite result (for resolutions of 32, 64, 128 etc., microns) using histology and diffusion MRI. It would be useful if the authors could comment on this discrepancy.

- The generation of ODFs based on the SLI data is innovative and very compelling. In the Methods section regarding ODFs, the authors describe the third set of 20 orientations used to approximate the dMRI signal off-plane. I found it hard to understand why adding this off-plane signal was necessary. It would be useful to clarify this point.

- Running tractography on this data is a useful demonstration, however I would advise to state explicitly possible concerns regarding running tractography on such data. For example, using very high resolutions, a small step size might cause the resulting streamline to "jump" from one fiber tract to the other (for example in the corpus callosum, jumping between the two populations that are separated by 10 degrees). In addition, it might be useful to demonstrate whether specific known tracts can be segmented from this whole-slice tractogram.

- Supp. Fig. 4 suggests that deparaffinization, dramatically reduces the ability to discern fiber orientations in the cortical gray matter. Please comment on this. In addition, this might be an interesting point of comparison with Nissl-ST, since in Nissl-ST, the observed orientations in the cortical gray matter are often perpendicular to the cortical sheet, as Nissl-ST picks up the columnar organization of the cortex more than the existence of (possibly unmyelinated) fibers that run through the cortex.

(Remarks on code availability)

Reviewer #2

(Remarks to the Author)

In this manuscript, the authors reported a fiber mapping method called ComSLI. ComSLI works by illuminating tissue sections from various angles using an LED. Local fibers scatter light differently depending on the relative orientation between the illumination angle and the local fiber orientation. Rich information can be derived, including fiber orientation and its distribution. ComSLI is capable of discerning multiple populations of fibers. Unlike polarization-based techniques that rely on birefringence, SLI is based on scattering, which directly ties to the local fiber organization and makes it ideal for characterizing tissues with weak or even no birefringence.

ComSLI itself, however, is not considered novel by this reviewer as ComSLI only represents an incremental improvement and is an extension of their SLI techniques reported earlier. The authors may want to compare ComSLI with their previously reported SLI techniques to highlight its novelty.

The strength of this study lies in the extensive demonstration of the applicability of ComSLI to tissue sections prepared with different protocols, stains, thicknesses, and storage periods, which makes ComSLI a highly relevant technique not only in neural science but also in other fields. The results are well presented in a rigorous way and support the conclusions from the study.

Light scattering has been implemented in various ways to probe the organization of local structures. The authors may want to expand the comparison between ComSLI and those techniques, such as SALS [Sacks, et al 1997].

Fig. 1G shows the intensity plots of one and two fiber populations, and Fig. 1I shows certain regions that have more than two fiber populations. It is not clear how to determine the fiber characteristics, orientation and distribution, for more than two populations.

As a transmission-based technique, how does the thickness of the tissue section affect the quantification accuracy?

Raw images were acquired at relatively large rotational steps. Was interpolation or any type of curve fitting used to quantify the fiber orientation? How was the rotational step determined?

(Remarks on code availability)

Version 2:

Reviewer comments:

Reviewer #1

(Remarks to the Author)

I would like to thank the authors for addressing the concerns I have raised during the first round of reviews, especially regarding the distinction between the microstructural and apparent fiber crossing at different resolutions. I find the current version of the manuscript clearer and strongly support its publication.

(Remarks on code availability)

Reviewer #2

(Remarks to the Author)

The authors have successfully addressed the concerns from this reviewer in the revised manuscript. Thank you!

(Remarks on code availability)

Point-by-Point Response to Reviewers' Comments

We would like to thank both reviewers for their constructive comments which helped us to further strengthen our manuscript. Following the reviewers' suggestions, we have thoroughly revised the manuscript as outlined below. As suggested by Reviewer #1, we have clarified our analysis of microscopic fiber crossings, added an analysis of apparent fiber crossings (Supplementary Fig. 1B), improved our description of ODF generation, discussed the effects of deparaffinization on gray-matter signals, and described the limitations of tractography algorithms. Following the suggestions from Reviewer #2, we have compared ComSLI to other light-scattering techniques like small-angle light scattering, clarified the analysis of more than two fiber populations and the determination of the azimuthal rotational step size, and added a paragraph to discuss the effects of different tissue thicknesses. In addition, we have shortened the Abstract to meet the formatting guidelines of revised manuscripts.

All changes made to the original manuscript and supplementary are marked as tracked changes in the first attached Related Manuscript File ("*Manuscript_Supplementary_highlighted-changes.pdf*"). Please see below for a detailed point-by-point response to all reviewers' comments and the corresponding changes. The given line numbers refer to the revised manuscript with accepted changes (see attached Article File).

We hope that we were able to address all points raised by the reviewers adequately, and that our revised manuscript is now suitable for publication in *Nature Communications*.

Reviewer #1

Georgiadis and colleagues present a method for measuring axonal (and other fibers) orientations based on SLI with micrometer resolution. Their method uses a novel setup to improve upon existing SLI-based methods and provide orientation data from different tissue preparations, including formalin-fixed paraffin-embedded sections. In addition, they provide an interesting usage of microstructure-informed orientation to create fiber orientation distributions (ODFs) similar to those used in the field of tractography. Importantly, they provide solid evidence that this new method is useful for studying microstructural changes in different pathological cases, and provide important comparisons with existing methods, showing the superiority of ComSLI compared with existing techniques. The fact that this technique is relatively cheap and easy to implement across various preparation procedures, both for stained and unstained samples, will make it of high interest to the neuroscientific and neuroimaging community. The manuscript is well-written and I strongly recommend its publication.

However, I believe that the following minor points should be addressed prior to publication.

- 1. The authors analyzed the prevalence of crossing fibers as a function of spatial resolution. Please highlight in this section that you are referring to in-plane orientations.***

We thank the reviewer for this comment. We are now mentioning both in the Results and in the Discussion as well as multiple times in the caption of Figure 1 that we are referring to *in-plane* fiber orientations.

Results (lines 150-153):

*This reveals that only ~7% of pixels at the original 7 μ m resolution contain two or more crossing **in-plane** fiber populations at the microscopic level, but this rises to 87% and 95% of pixels containing microscopic crossings for 500 μ m and 1mm pixel resolution, respectively (Supplementary Fig. 1A).*

Discussion (line 436):

*... only 7% of 7 μ m white matter pixels showed **in-plane** crossings, ...*

Figure 1 caption:

... (A) A light beam impinging on a fiber bundle scatters predominantly perpendicular to the fibers, resulting in paired peaks in the azimuthal intensity profile, $I(\varphi)$, whose midline indicates the **in-plane** fiber orientation. ...

... (C) ... The **in-plane** fiber orientations are shown as multi-colored pixels ...

... (G) Azimuthal intensity signals from pixels with single and two **in-plane** crossing fiber populations ...

... (I) Map of the number of **in-plane** fiber populations in each $7\mu\text{m}$ pixel of the white matter. ...

- 2. The same analysis shows that the number of crossing fibers increases with increasing “voxel” size (Supp. Fig. 1). This contrasts with previous work by Schilling et al. (“Can increased spatial resolution solve the crossing fiber problem for diffusion MRI?”, NMR Biomed, 2017) that showed the opposite result (for resolutions of 32, 64, 128 etc., microns) using histology and diffusion MRI. It would be useful if the authors could comment on this discrepancy.**

We thank the reviewer for the great comment and for highlighting the relevant work from Schilling et al. (2017). These authors found indeed that the prevalence of crossing fibers increases with increasing spatial resolution. This in principle agrees with our finding that higher resolutions uncover crossings which would otherwise be missed in lower resolutions. The apparent “contradiction” stems from the fact that our study was focusing on the number of *microscopic* crossings and not *apparent* crossings at lower resolutions: As noted in the Methods, for each resolution, we had marked a “pixel” as containing crossings if its corresponding area (e.g., $50 \times 50 \mu\text{m}^2$) contained microscopic fiber crossings at the highest resolution. For example, if a 7×7 kernel of 7-micrometer ComSLI pixels (corresponding to an area of $\sim 50 \times 50 \mu\text{m}^2$) contained microscopic pixels with two (or three) fiber populations, indicating microscopic crossings, the area was displayed in yellow (or red), indicating that crossing fibers are present within this kernel.

We have now performed additional analysis to directly study the effect of apparent crossings in changing resolutions: we have downsampled our original micron-resolution images to the same set of lower resolutions, and performed SLIX analysis on them to derive the fiber orientations in lower resolutions. As expected, and in agreement with Schilling et al., we find that less crossings are detected at lower resolutions, as the signal from the multiple fiber orientations at higher resolutions gets averaged across multiple pixels, which expectedly results in a loss of information.

We have added these new findings in Supplementary Fig. 1B (see below). In addition, we have added a more detailed description to the manuscript to clarify our analysis (for both microscopic and apparent crossings), and compared our results to the study from Schilling et al. (new reference no. 41).

Results (lines 148-155):

For instance, one can calculate how many brain pixels contain fiber crossings at the ComSLI resolution (Fig. 1I), and at multiple other resolutions common for MRI scanning (Supplementary Fig. 1). This reveals that only ~7% of pixels at the original $7\mu\text{m}$ resolution contain two or more crossing in-plane fiber populations, but this rises to 87% and 95% of pixels containing microscopic crossings for $500\mu\text{m}$ and 1mm pixel resolution, respectively (Supplementary Fig. 1A). Conversely, the number of apparent, detectable crossings decreases at lower resolutions (Supplementary Fig. 1B), possibly as a result of the averaging of the signal and loss of orientation information at larger pixel sizes.

Discussion (lines 434-443):

This included calculating the number of crossing fibers in the brain (Supplementary Fig. 1). When studying the presence of microscopic crossings at multiple resolutions, only 7% of $7\mu\text{m}$ white matter pixels showed in-plane crossings, which rises to ~95% of 1mm pixels containing microscopic crossings (Supplementary Fig. 1A). This is higher than previous estimates³⁰, highlighting the fact that lower resolutions underestimate the number of crossings that are actually present and can be revealed with higher (microscopic in our case) resolutions. This effect is shown in Supplementary Fig. 1B, where we study detectability of crossings by resolution for the same pixel sizes, finding lower percentage of detectable crossings in larger pixels. This is in line with findings from a previous MRI and histology study⁴¹, where lower resolution imaging uncovered lower number of crossings.

Supplementary Fig. 1 | Number of fiber populations at different pixel sizes. (A) Presence of microscopic crossings. Maps of the number of *microscopic* fiber populations with different orientations per pixel, for various pixel sizes, starting from single pixels at the ComSLI resolution for this experiment (7 μm), then pixel kernels of 7x7 *microscopic* pixels (~50 μm), 14x14 pixels (~100 μm), 71x71 pixels (~500 μm), and 143x143 pixels (~1mm). *The lower resolution 50 μm -1mm pixels were marked as having multiple fiber populations if their kernel contained microscopic (7 μm) pixels with multiple fiber populations.* **(B) Detectability of crossings by resolution.** For apparent crossings, the original ComSLI image data were downsampled to the respective resolution, and SLIX was used to identify fiber orientations/crossings.

Methods (new subsection, lines 727-744):

Number of crossings

To study the number of fiber crossings, as presented in Fig. 1I and Supplementary Fig. 1, we used the output of the orientations of the SLIX software³³. Briefly, SLIX determines the number of orientations in each microscopic pixel by finding the number of peak pairs in each pixel's intensity-vs-angle plot (Fig. 1G) that correspond to fiber populations with distinct orientations^{24,33} (up to 3 fiber populations per microscopic pixel). Each peak pair indicates one fiber population, i.e., an intensity profile with one peak pair (2 peaks) corresponds to a single fiber population, two peak pairs (4 peaks) correspond to two fiber populations, and 3 peak pairs (6 peaks) to 3 fiber populations (the maximum that can robustly and accurately be resolved in a micrometer-sized pixel). For the multi-resolution analysis in Supplementary Fig. 1A, we considered the number of microscopic crossings in each kernel making up the lower resolution "pixels". For example, if a 7x7 kernel of 7-micrometer ComSLI pixels (corresponding to an area of $\sim 50 \times 50 \mu\text{m}^2$) contained microscopic pixels with two (or three) fiber populations, indicating microscopic crossings, the lower resolution "pixel" was displayed in yellow (or red), indicating that crossing fibers are present within this kernel. For the analysis of apparent crossings in Supplementary Fig. 1B, we downsampled the original ComSLI images to the respective resolutions, and employed SLIX to retrieve fiber orientations per pixel in the downsampled datasets, from which the number of orientations per pixel was determined.

- 3. The generation of ODFs based on the SLI data is innovative and very compelling. In the Methods section regarding ODFs, the authors describe the third set of 20 orientations used to approximate the dMRI signal off-plane. I found it hard to understand why adding this off-plane signal was necessary. It would be useful to clarify this point.***

We are glad that the reviewer appreciates the presented generation of ODFs based on the ComSLI data, and thank the reviewer for this question. In adding an off-plane diffusion signal component to the in-plane one computed from the ComSLI orientations, we aimed to approximate a realistic diffusion signal which at each point would be composed by a convolution of the fiber orientations and the 3D fiber response. The first step in the most common diffusion tractography processes (which we used) is to estimate the fiber response using a spherical deconvolution. Therefore, we wanted to simulate a realistic signal including a fiber response with a sharp peak along the axis and a quick fall-off, as suggested by Tournier et al. (Tournier et al., "Direct estimation of the fiber orientation density function from diffusion-weighted MRI data using spherical deconvolution", NeuroImage, 2004) which is the basis of the MRtrix3 tractography algorithm. We have added this reference and clarified this point in the Methods section.

Methods (lines 769-773):

This aimed to approximate a realistic diffusion MRI signal and fiber distribution off-plane, where the signal loss at 20° off the detected in-plane axon orientation is 20% of the signal loss along the axons, which reflects a fiber response with a sharp peak along the axis and a quick off-axis fall-off, as described by Tournier et al.⁷¹, and is expected for the following tractography steps that include estimating the fiber response.

- 4. Running tractography on this data is a useful demonstration, however I would advise to state explicitly possible concerns regarding running tractography on such data. For example, using very high resolutions, a small step size might cause the resulting streamline to "jump" from one fiber tract to the other (for example in the corpus callosum, jumping between the two populations that are separated by 10 degrees). In addition, it might be useful to demonstrate whether specific known tracts can be segmented from this whole-slice tractogram.***

It is true that in high-resolution data, a small step size might cause streamline algorithms to "jump" between neighbored tracts. We would like to mention that we did not perform tractography on the micron-resolution ComSLI data, but on the lower-resolution microstructure-informed kernel data (20 or 50 μm) as defined by our μFODs . However, these are likely still high resolution for typical MRI tractography algorithms. We have added this valid remark to the Discussion.

Given the low thickness of the histology sections, we believe that tractography from ComSLI data can nicely visualize short fiber tracts reflecting the overall anatomy of the section, but it would be hard to segment tracts based on a whole-slice tractogram because most fiber tracts have a 3-dimensional course that extends far beyond the geometrical limits of a 2D histology section. For segmenting individual fiber tracts, it would be necessary to measure and carefully register several consecutive tissue sections before applying fiber tractography algorithms, which is something we plan to do in subsequent studies; future work will look into fiber tractography on such 3D-ComSLI data sets and also compare the findings to previous tracing studies – complementing, e.g., existing brain atlases.

Following the reviewer's suggestion, we have added the above considerations to the Discussion and explicitly stated the limitations when running tractography on the presented ComSLI data.

Discussion (lines 556-565):

Regarding tractography applications, the low section thickness only allows to follow underlying anatomic tracts over a short course, limiting the study of single sections to short-range fiber tracts; analysis of consecutive sections of a 3D specimen can enable studying longer-range tracts and will be the topic of next studies. Also, the resulting tractograms should be interpreted with caution, as the employed tractography methods were developed for dMRI which has a lower resolution than ComSLI, and the microscopic resolution can result in fiber streamlines "jumping" between neighboring tracts with small angle separations; refinement of tractography algorithms for ComSLI and detailed anatomical validation using tracer studies will be the subject of future work.

- 5. *Supp. Fig. 4 suggests that deparaffinization, dramatically reduces the ability to discern fiber orientations in the cortical gray matter. Please comment on this. In addition, this might be an interesting point of comparison with Nissl-ST, since in Nissl-ST, the observed orientations in the cortical gray matter are often perpendicular to the cortical sheet, as Nissl-ST picks up the columnar organization of the cortex more than the existence of (possibly unmyelinated) fibers that run through the cortex.***

We thank the reviewer for this remark. It is true that deparaffinization reduces the ability to discern fiber orientations in the cortex (as shown in Supplementary Figure 4). While still in paraffin, cortical fibers are better discernible possibly aided by the fact that paraffin has a higher refractive index which increases the scattering of unmyelinated fibers in the cortex. After deparaffinization, this refractive index difference is reduced. If one also takes into account that in the shown deparaffinized step (Supplementary Figure C,D) the tissue has become almost fully transparent after the use of xylene, the signal-to-noise ratio in the cortex becomes so small that it is hard to detect small refractive index differences (on which light scattering relies) and hence to extract reliable fiber orientations in the gray matter at this specific stage.

Regarding Nissl-ST, the obtained fiber orientations in the cortical gray matter (e.g., in Figure 4A') show indeed mostly orientations perpendicular to the cortical sheet, caused by the columnar cellular organization of the cortex. We refrained from making a direct method comparison in cortical gray matter given that the authors of the Nissl-ST manuscript present their method as being able to retrieve orientations in white matter only, possibly aware of the above mentioned limitations in the gray matter.

Following the reviewer's suggestion, we have added these considerations to the manuscript.

Results (lines 350-351):

Gray matter comparison was not pursued given Nissl-ST's focus on white matter²⁸.

Discussion (lines 459-469):

Although ComSLI can provide fiber orientations in white and gray matter at different steps of sample preparation, it seems to have low sensitivity to gray matter fiber orientations in the almost transparent, deparaffinized sections (Supplementary Fig. 4). This can be explained by a combination of effects: first, paraffin has a higher refractive index than axons which can increase the scattering of (unmyelinated) fibers in the cortex. Second, treatment with xylene to remove paraffin results in an almost complete loss of refractive in-

dex differences, which is the basis of the scattered light signal²⁹, rendering the sample almost fully transparent (Supplementary Fig. 4C,D). Although ComSLI can still retrieve signal from the white matter, it is unable to reliably discern gray matter fiber orientations at this specific stage due to very low signal-to-noise levels. Staining appears to partly restore the refractive index differences and enable retrieving some fiber orientations in cortical gray matter.

Reviewer #2

In this manuscript, the authors reported a fiber mapping method called ComSLI. ComSLI works by illuminating tissue sections from various angles using an LED. Local fibers scatter light differently depending on the relative orientation between the illumination angle and the local fiber orientation. Rich information can be derived, including fiber orientation and its distribution. ComSLI is capable of discerning multiple populations of fibers. Unlike polarization-based techniques that rely on birefringence, SLI is based on scattering, which directly ties to the local fiber organization and makes it ideal for characterizing tissues with weak or even no birefringence.

The strength of this study lies in the extensive demonstration of the applicability of ComSLI to tissue sections prepared with different protocols, stains, thicknesses, and storage periods, which makes ComSLI a highly relevant technique not only in neural science but also in other fields. The results are well presented in a rigorous way and support the conclusions from the study.

- 1. ComSLI itself, however, is not considered novel by this reviewer as ComSLI only represents an incremental improvement and is an extension of their SLI techniques reported earlier. The authors may want to compare ComSLI with their previously reported SLI techniques to highlight its novelty.***

We agree that ComSLI builds on the existing SLI technique, and we therefore refrained from claiming that we present a completely new technique, highlighting SLI and past applications in the Introduction and Discussion. However, we believe our study is still a significant advance to the existing method as it presents multiple novel aspects (as pointed out in the Introduction and Discussion): We use a novel setup for improved signal-to-noise ratio (rotating directed high-power LED light instead of LED display, please see Figure below) which makes it easier to realize ComSLI in other laboratories and enables the advanced studies presented in the manuscript. While SLI has only been applied to unstained and fixed cryo-sections of smaller brain tissue samples, ComSLI allowed us to reveal fiber structures in all kinds of differently prepared samples (FFPE, different stains, unfixed fresh-frozen), in whole human brain and pathological samples, as well as in non-brain tissues. In addition, we present a novel advanced analysis framework, including microstructure-informed fiber orientation distributions and tractograms. As pointed out by the reviewer, the main strength of our study is the detailed and quantitative demonstration of the applicability of ComSLI across tissues and preparation protocols, using new analysis tools, making ComSLI a highly relevant technique for neuroscience and beyond.

Comparison of fiber orientation maps from the SLI setup (left) and ComSLI setup (right), for a region of the silver-stained BigBrain section (main Fig. 1C, between rectangles D and E). The ComSLI setup uses a directed high-power LED light instead of an LED display and retrieves more robust fiber orientations with 125ms exposure time and no gain, while the SLI setup needed an exposure time of 5 seconds and a gain of 5 to reach an inferior result.

2. Light scattering has been implemented in various ways to probe the organization of local structures. The authors may want to expand the comparison between ComSLI and those techniques, such as SALS [Sacks, et al 1997].

It is true that techniques like small-angle light scattering (SALS) also use the scattering of light to probe structural organization. In contrast to ComSLI, they raster-scan the tissue with a pencil beam of hundreds of micrometers which makes it impossible to assess fiber organization over large fields of view at μm -resolution. It is a great suggestion to discuss the ComSLI results also in comparison to those techniques. We thank the reviewer for this comment, and have included the suggested reference (Sacks et al., 1997) as well as a comparison/reference to other light-scattering techniques in the Introduction and Discussion. Regarding experimental comparison to such pencil-beam light scattering methods, we have previously directly compared SLI to coherent Fourier scatterometry (Menzel & Pereira, "Coherent Fourier scatterometry reveals nerve fiber crossings in the brain", *Biomedical Optics Express* 11, 2020), which is very similar in principle with SALS but extends to higher scattering angles (including those probed by SLI and ComSLI), and have found an excellent agreement between the two techniques.

Introduction (lines 46-49):

Small-angle light scattering (SALS) similarly uses a pencil beam of visible light photons to provide local fiber orientations^{12,13}, but cannot reach micrometer resolution and requires long scans and dedicated laser illumination and detection equipment¹².

Discussion (lines 526-534):

Scattering of light to study fiber orientations has been previously also used in methods such as small-angle light scattering (SALS), which captures light scattering from tissues by raster-scanning them using a pencil laser beam of hundreds of micrometers^{12,65}. In contrast, ComSLI captures a centimeter-wide field of view within seconds with micrometer resolution and without the need for specialized laser beam equipment. Similar to SALS, we have previously applied coherent Fourier scatterometry using a laser beam and capturing light at both low and wide angles to study the physics of light scattering from brain tissues⁶⁶, and we found excellent agreement of the overall fiber orientations with those determined by scattered light imaging in the same tissue sections²⁴.

3. Fig. 1G shows the intensity plots of one and two fiber populations, and Fig. 1I shows certain regions that have more than two fiber populations. It is not clear how to determine the fiber characteristics, orientation and distribution, for more than two populations.

We thank the reviewer for this comment. The intensity plots in Figure 1G show example pixels with one fiber population (top, corpus callosum) and two fiber populations (bottom, corona radiata); the number of fiber populations was determined by the SLIX software (see Methods) from the number of prominent peak pairs in each pixel's intensity-vs-angle plot. Each peak pair indicates one fiber population, i.e., an intensity profile with three peak pairs (six peaks) would correspond to three fiber populations (see Figure below). In Figure 1I, the determined number of fiber populations is shown for each ComSLI image pixel. For each microscopic pixel, the fiber orientation information is retrieved, but not the distribution, since the number of fibers contained in it is too low to allow a robust distribution determination. The fiber orientation distributions are calculated when pooling multiple microscopic pixels into kernels, shown in Fig. 1H as polar plots. We have added some sentences to better describe how fiber characteristics were computed in the case of three fiber populations.

Methods (lines 729-736):

Briefly, SLIX determines the number of orientations in each microscopic pixel by finding the number of peak pairs in each pixel's intensity-vs-angle plot (Fig. 1G) that correspond to fiber populations with distinct orientations^{24,33} (up to 3 fiber populations per microscopic pixel). Each peak pair indicates one fiber population, i.e., an intensity profile with one peak pair (2 peaks) corresponds to a single fiber population, two peak pairs (4

peaks) correspond to two fiber populations, and 3 peak pairs (6 peaks) to 3 fiber populations (the maximum that can robustly and accurately be resolved in a micrometer-sized pixel).

Three crossing fiber layers (left) and corresponding intensity-angle plot (right) for the location indicated by the black circle. The orientation from each fiber population appears in the plot as a peak pair, with three fiber populations corresponding to six peaks in the plot. Blue lines correspond to results from coherent Fourier scatterometry, orange lines to results from scattered light imaging.

4. As a transmission-based technique, how does the thickness of the tissue section affect the quantification accuracy?

This is a great question that has also concerned us a lot. Tissue thickness affects the light attenuation (scattering and absorption) and thus the strength of the measured scattering signal in two ways: With a larger tissue thickness, there are more fiber layers that scatter light, but the light is also attenuated more.

We have tried to assess this effect in multiple ways: First, indirectly, by including multiple tissue thicknesses in the study, where FFPE sections have been as thin as 4 μ m (all hippocampal and pathological human sections) to 20 μ m thick for the (cell-body stained) BigBrain sections. For unstained sections without paraffin embedding, brain tissue sections of 60 μ m thickness have successfully been analyzed (Menzel et al., NeuroImage 233, 2021 & Front Neuroanat 15, 2021), as well as vibratome sections of around 100 μ m thickness (Menzel et al., eLife 12, 2023).

We have also tried to directly assess the effect of thickness: As shown in Supplementary Figure 13, it is possible to accurately and reproducibly reveal fiber orientations across different section thicknesses (demonstrated for tongue muscle tissue of 4–15 μ m thickness).

As for thicker samples, or samples that cannot be sectioned, future work will look into the realization of back-scattered light imaging to probe tissue surfaces (independent of tissue thickness).

Following the reviewer's suggestion, we have added a more detailed discussion to the manuscript how the section thickness affects the quantification accuracy.

Discussion (lines 535-550):

As a light microscopy transmission-based technique, ComSLI analyzes histological tissue sections, and the tissue thickness might affect measurement results: with a higher tissue thickness, there are more fiber layers that scatter light (increasing the signal), but the light is also attenuated more (decreasing the signal). In previous studies, through-scattered light signals could be retrieved from sections as thick as 500 μ m using laser sources¹². Here, we tried to assess the effect of tissue thickness in multiple ways: First, indirectly, by studying multiple commonly used tissue thicknesses, e.g., FFPE sections ranging from 4 μ m (all hippocampal and pathological human sections) to 20 μ m thick for the (cell-body stained) BigBrain sections. Second, we tried to directly assess the effect of thickness: As shown in Supplementary Figure 13, it is possible to accurately and reproducibly reveal fiber orientations across different section thicknesses (demonstrated for tongue muscle tissue of 4–15 μ m thickness). For unstained sections without paraffin embedding, brain tissue sections of 60 μ m thickness have successfully been analyzed^{24,25}, as well as vibratome sections of around 100 μ m thickness²⁶. As for thicker samples, or samples that cannot be sectioned, future work will investigate the realization of back-scattered light imaging (instead of transmission mode as shown here) to probe tissue surfaces independent of tissue thickness.

5. Raw images were acquired at relatively large rotational steps. Was interpolation or any type of curve fitting used to quantify the fiber orientation? How was the rotational step determined?

As can be seen in the raw data from single pixels in Fig. 1G, the intensity peaks span a range of tens of degrees, so our rotational steps allow us to capture multiple points of the curve. In previous SLI studies (Menzel et al., NeuroImage 233, 2021), a rotational step of 15° was found as a good compromise of measurement time and accuracy; discretization artifacts were accounted for by computing the center of gravity of the peak tips enabling to accurately capture the peak position, as validated with coherent Fourier scattering (see also above figure 3 from that study). Samples in the current manuscript were measured either with the same step (15°) or with a smaller step of 10° and with low-pass frequency (Fourier) filtering to smoothen the curves (as described in the Methods), with parameters carefully selected as part of the previous work (Menzel et al., Front. Neuroanat. 15, 2021). We have added the following text to the Methods section to refer to this work validating our parameters.

Methods (line 725-726):

Previous studies have validated the step size as well as the filtering parameters used²⁴⁻²⁵.